# FiX: Introducing Fine-grained Forget Gate into Softmax Attention

**Runzhong Li** [1 2]  **Renjie Liu** [1 †]  **Qing Li** [2]  **Bo Tang** [1]

Causal softmax attention is the algorithmic foundation of modern large language models. Inspired by linear attention, recent work has sought to enhance it by incorporating forget gates. However, these efforts, such as FoX, have been limited to coarse, scalar gates. While fine-grained, element-wise gates are shown to be more effective than scalar ones in linear attention, their direct integration into softmax attention is non-trivial due to algebraic constraints. In this work, we introduce *Fine-grained Forgetting Transformer* (*FiX*), a novel architecture that successfully enables element-wise forget gates in softmax attention. Our core insight is that the softmax denominator becomes mathematically redundant under a subsequent RMSNorm layer, allowing us to reformulate the forgetting mechanism as a direct element-wise multiplication on the value vectors. This formulation makes FiX the first positional encoding applied to value-output (VO) pairs, designed to be complementary to existing query-key (QK) encodings like RoPE. We systematically address implementation challenges including numerical precision, computational efficiency, and inference memory consumption. Extensive experiments show *FiX* achieves lower training loss and superior performance on both short-text common sense benchmarks and long-context tasks, opening a new path for building more powerful transformers. Our code is available at https://github.com/AlayaDB-AI/FiX.

## 1. Introduction

Despite the surge of research into efficient alternatives like recurrent sequence models (Yang et al., 2025b; Zhang et al., 2025b) to mitigate the quadratic complexity of standard attention, they frequently fall short in long-context scenarios due to the inherent constraints of their fixed-size recurrent states (Wen et al., 2025). Consequently, causal softmax attention (Vaswani et al., 2017) remains the unrivaled algorithmic foundation and an essential component for modern

large language models (LLMs) (DeepSeek-AI et al., 2025; Yang et al., 2025a; Team et al., 2025). To further enhance its capabilities, recent work such as Forgetting Transformers (FoX) (Lin et al., 2025) has integrated forget gate mechanisms (Hochreiter & Schmidhuber, 1997; Sun et al., 2023) from recurrent sequence models into the softmax attention, enabling models to selectively discard historical information based on context. However, these efforts have been limited to coarse, scalar gates that treat all feature dimensions with a uniform decay rate. This lack of fine-grained, element-wise control is a significant limitation, as such mechanisms have been shown to effectively improve modern linear attention models, e.g., Gated Linear Attention (Yang et al., 2024) and Kimi Delta Attention (Zhang et al., 2025b).

However, translating this success from linear attention to softmax attention is non-trivial, since directly extending FoX to support fine-grained gating is infeasible. Specifically, in FoX, the forgetting mechanism is applied as a bias term $D_{i,j}$ added to the attention logit $q_i k_j^\top$. Since the dot product of query and key vectors yields a scalar, it is algebraically impossible to add a vector-valued bias (namely a multi-dimensional forget gate) to this scalar logit.

To enable vectorized forget gates in softmax attention, we analyze the behavior of softmax attention when followed by a Root Mean Square Layer Normalization (RMSNorm) (Zhang & Sennrich, 2019), a standard component in modern LLM architectures (Yang et al., 2025a). Interestingly, we find that under RMSNorm, the normalization denominator of the softmax operation becomes redundant. This insight suggests that the mathematical effect of a forget gate within the softmax mechanism is functionally equivalent to applying cumulative forget gates to the value vectors.

Under this observation, we propose the *Fine-grained Forgetting Transformer (FiX)*. By decoupling the forgetting mechanism from the attention scores, *FiX* achieves the fine-grained control necessary for handling complex long-context dependencies without sacrificing the foundational strengths of the transformer architecture. Moreover, this fine-grained forget gate also serves as V-O position embeddings, making it complementary with existing Q-K positional encoding like RoPE (Su et al., 2024). To our understanding, *FiX* is the first positional encoding on the V-O vectors.

For hardware-efficient training, we implement *Flash Fine-*

†Work done during internship at AlayaDB. [1]Department of Computer Science and Engineering, Southern University of Science and Technology, Shenzhen, China [2]Department of Computing, The Hong Kong Polytechnic University, Hong Kong, China. Correspondence to: Bo Tang <tangb3@sustech.edu.cn>, Qing Li <csqli@comp.polyu.edu.hk>.

*Proceedings of the 43rd International Conference on Machine Learning*, Seoul, South Korea. PMLR 306, 2026. Copyright 2026 by the author(s).

*grained Attention*, a FlashAttention-like (Dao, 2024) algorithm designed for *FiX*. This algorithm combines the rescaling trick from Section 3.2.3 with the tiling strategy of FlashAttention to balance numerical stability and efficiency. To improve the numerical stability of RMSNorm in our setting ($\epsilon = 10^{-30}$ in O-Norm), we fuse the *FiX*, O-Norm, and the subsequent linear layer into a single operator. For inference, we introduce a memory-saving cache to balance efficiency and memory usage, which integrates seamlessly with PagedAttention (Kwon et al., 2023). On moderate-scale language modeling with 760M-parameter transformers, our experiments show that *FiX* outperforms baselines such as RoPE and FoX on various tasks. Moreover, *FiX* can work with RoPE to get further improvements.

## 2. Background and Motivation

This section briefly reviews several key attention variants, identifies the main obstacle for applying a fine-grained forget gate to softmax attention, and presents the key observation that motivates *FiX*.

### 2.1. Softmax Attention and Linear Attention

In causal sequence modeling, each time step uses only historical information. Within this framework, an attention layer takes a sequence of input vectors $x_i$ with $i \in \{1, \ldots, N\}$, where $N$ is the sequence length and $i$ is the time step. All other sequences are projected from $x_i$, for example $q_i = x_i W_q$, $k_i = x_i W_k$, and $v_i = x_i W_v$. The layer then produces a sequence of outputs $o_i$.

Standard causal softmax attention is defined as:

$$o_i = \frac{\sum_{j=1}^{i} \exp\left(q_i k_j^{\mathrm{T}}\right) v_j}{\sum_{j=1}^{i} \exp\left(q_i k_j^{\mathrm{T}}\right)}. \tag{1}$$

We call this the *parallel form* following (Yang et al., 2024) as each $o_i$ can be computed independently. Then, replacing the term $\exp\left(q k^{\mathrm{T}}\right)$ with a kernel $\phi\left(q\right)\phi\left(k\right)^{\mathrm{T}}$, where $\phi\left(\cdot\right)$ is a non-negative feature map, yields the parallel form of a linear attention (Katharopoulos et al., 2020):

$$\begin{aligned} o_i &= \frac{\sum_{j=1}^{i} \phi\left(q_i\right)\phi\left(k_j\right)^{\mathrm{T}} v_j}{\sum_{j=1}^{i} \phi\left(q_i\right)\phi\left(k_j\right)^{\mathrm{T}}} \\ &= \frac{\phi\left(q_i\right)\sum_{j=1}^{i} \phi\left(k_j\right)^{\mathrm{T}} v_j}{\phi\left(q_i\right)\sum_{j=1}^{i} \phi\left(k_j\right)^{\mathrm{T}}}. \end{aligned} \tag{2}$$

This expression can be written in *recurrent form* equivalently as an RNN, where $S_i$ and $z_i$ are fixed-size RNN states:

$$\begin{aligned} S_i &= S_{i-1} + \phi\left(k_i\right)^{\mathrm{T}} v_i, \\ z_i &= z_{i-1} + \phi\left(k_i\right), \\ o_i &= \frac{q_i S_i}{q_i z_i}. \end{aligned} \tag{3}$$

### 2.2. Forget Gate and Forgetting Attention (FoX)

The forget gate is a long-established component of RNNs, e.g., LSTM (Hochreiter & Schmidhuber, 1997). Here we define a data-dependent forget gate as a scalar $f_i = \text{sigmoid}\left(x_i w_f^{\mathrm{T}} + b\right)$. Applying this gate to the recurrent form of the above linear attention yields:

$$\begin{aligned} S_i &= f_i S_{i-1} + \phi\left(k_i\right)^{\mathrm{T}} v_i, \\ z_i &= f_i z_{i-1} + \phi\left(k_i\right). \end{aligned} \tag{4}$$

This is equivalent to the following parallel form:

$$o_i = \frac{\sum_{j=1}^{i} \left(\prod_{t=j+1}^{i} f_t\right) \phi\left(q_i\right)\phi\left(k_j\right)^{\mathrm{T}} v_j}{\sum_{j=1}^{i} \left(\prod_{t=j+1}^{i} f_t\right) \phi\left(q_i\right)\phi\left(k_j\right)^{\mathrm{T}}}. \tag{5}$$

Substituting the feature-map kernel $\phi\left(q\right)\phi\left(k\right)^{\mathrm{T}}$ with the original exponential kernel $\exp\left(q k^{\mathrm{T}}\right)$ used in softmax attention gives the forgetting attention mechanism in FoX:

$$o_i = \frac{\sum_{j=1}^{i} \left(\prod_{t=j+1}^{i} f_t\right) \exp\left(q_i k_j^{\mathrm{T}}\right) v_j}{\sum_{j=1}^{i} \left(\prod_{t=j+1}^{i} f_t\right) \exp\left(q_i k_j^{\mathrm{T}}\right)}. \tag{6}$$

### 2.3. Fine-grained Forget Gate

Many linear-attention variants employ a fine-grained forget gate, which has been shown to outperform a scalar gate. This data-dependent fine-grained forget gate can be defined as the vector $f_i = \text{sigmoid}\left(x_i W_f + b\right)$. Applying this gate to the recurrent form of the above linear attention gives:

$$\begin{aligned} S_i &= \text{diag}\left(f_i\right) S_{i-1} + \phi\left(k_i\right)^{\mathrm{T}} v_i, \\ z_i &= f_i \circ z_{i-1} + \phi\left(k_i\right), \end{aligned} \tag{7}$$

where $\circ$ denotes element-wise multiplication. Again, we can transform this recurrent form to the parallel form:

$$o_i = \frac{\sum_{j=1}^{i} \phi\left(q_i\right)\left(\prod_{t=j+1}^{i} \text{diag}\left(f_t\right)\right) \phi\left(k_j\right)^{\mathrm{T}} v_j}{\sum_{j=1}^{i} \phi\left(q_i\right)\left(\prod_{t=j+1}^{i} \text{diag}\left(f_t\right)\right) \phi\left(k_j\right)^{\mathrm{T}}}. \tag{8}$$

However, because now the kernel $\phi\left(q\right)\phi\left(k\right)^{\mathrm{T}}$ no longer appears explicitly, we cannot directly transfer this fine-grained forget gate to softmax attention using the same substitution that derives Equation 6 from Equation 5.

### 2.4. Motivation: Equivalence of QK-FoX and VO-FoX

FoX, which uses scalar forgetting gates in softmax attention, can be viewed as a type of positional encoding acting on $q$ and $k$. Defining $\gamma_i = \prod_{t=1}^{i} f_t$, $\gamma_{j \to i} = \prod_{t=j+1}^{i} f_t = \frac{\gamma_i}{\gamma_j}$,

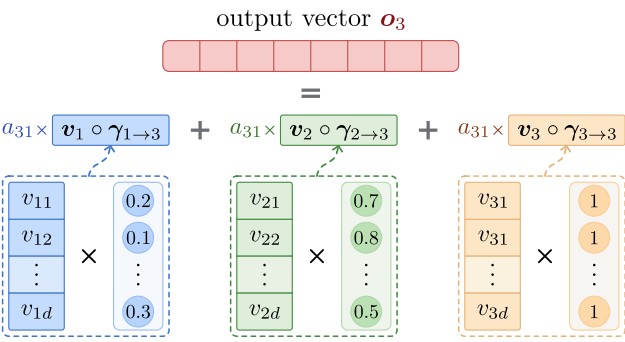

*Figure 1.* A conceptual illustration of *FiX*.

and $D_{ij} = \log \gamma_i - \log \gamma_j$, we can rewrite Equation 6 as:

$$
\begin{aligned}
\boldsymbol{o}_i &= \frac{\sum_{j=1}^{i} \frac{\gamma_i}{\gamma_j} \exp\left(\boldsymbol{q}_i \boldsymbol{k}_j^{\mathrm{T}}\right) \boldsymbol{v}_j}{\sum_{j=1}^{i} \frac{\gamma_i}{\gamma_j} \exp\left(\boldsymbol{q}_i \boldsymbol{k}_j^{\mathrm{T}}\right)}, \\
&= \frac{\sum_{j=1}^{i} \exp\left(\boldsymbol{q}_i \boldsymbol{k}_j^{\mathrm{T}} + D_{ij}\right) \boldsymbol{v}_j}{\sum_{j=1}^{i} \exp\left(\boldsymbol{q}_i \boldsymbol{k}_j^{\mathrm{T}} + D_{ij}\right)}.
\end{aligned}
\tag{9}
$$

We denote this form as the QK-decay version of FoX (QK-FoX). If the decay is omitted from the denominator, Equation 10 becomes a positional encoding on $\boldsymbol{v}$ and $\boldsymbol{o}$:

$$
\begin{aligned}
\boldsymbol{o}_i' &= \frac{\sum_{j=1}^{i} \frac{\gamma_i}{\gamma_j} \exp\left(\boldsymbol{q}_i \boldsymbol{k}_j^{\mathrm{T}}\right) \boldsymbol{v}_j}{\sum_{j=1}^{i} \exp\left(\boldsymbol{q}_i \boldsymbol{k}_j^{\mathrm{T}}\right)}, \\
\frac{\boldsymbol{o}_i'}{\gamma_i} &= \frac{\sum_{j=1}^{i} \exp\left(\boldsymbol{q}_i \boldsymbol{k}_j^{\mathrm{T}}\right) \frac{\boldsymbol{v}_j}{\gamma_j}}{\sum_{j=1}^{i} \exp\left(\boldsymbol{q}_i \boldsymbol{k}_j^{\mathrm{T}}\right)}.
\end{aligned}
\tag{10}
$$

We denote this form as the VO-decay version of FoX (VO-FoX). Now we consider how VO-FoX differs from QK-FoX (standard FoX) under RMSNorm (Zhang & Sennrich, 2019), a standard component in modern LLMs defined as:

$$
\mathrm{RMSNorm}\left(\boldsymbol{x}; \boldsymbol{w}\right)_i = \frac{x_i w_i}{\sqrt{\frac{\sum_{j=1}^{n} x_j^2}{n} + \epsilon}}.
\tag{11}
$$

Here $n$ is the dimension of the input vector $\boldsymbol{x}$ and $\epsilon$ is a small constant. In particular, when $\epsilon = 0$, the results of $\boldsymbol{o}$ and $\boldsymbol{o}'$ in Equation 9 and 10 are the same after RMSNorm:

$$
\mathrm{RMSNorm}\left(\boldsymbol{o}_i'\right) = \mathrm{RMSNorm}\left(\boldsymbol{o}_i\right).
\tag{12}
$$

As the attention output is always RMS-normalized before subsequent operations, this equivalence indicates that the VO-decay version of FoX is effectively the same as standard FoX under RMSNorm. Consequently, this observation motivates us to introduce a fine-grained forget gate into softmax attention via $\boldsymbol{v}$ and $\boldsymbol{o}$, rather than through $\boldsymbol{q}$ and $\boldsymbol{k}$.

## 3. Fine-grained Forgetting Transformer

Inspired by the key insight discussed in Section 2.4, we propose *Fine-grained Forgetting Transformer* (*FiX*). The core

of *FiX* is a causal softmax attention with fine-grained forget gates applied on $\boldsymbol{v}$ and $\boldsymbol{o}$, which is named *Fine-grained Forgetting Attention*.

At each time step the fine-grained forget gate is a vector $\boldsymbol{f}_i$ in which each element $\in (0, 1)$. We will discuss details about the specific generation of the forget gates in Section 3.1. We define the cumulative fine-grained forget gate $\boldsymbol{\gamma}_i = \prod_{j=1}^{i} \boldsymbol{f}_i$, where the multiplication is element-wise multiplication. Then the output of the *Fine-grained Forgetting Attention* can be formulated as:

$$
\begin{aligned}
\boldsymbol{o}_i &= \frac{\sum_{j=1}^{i} \exp\left(\boldsymbol{q}_i \boldsymbol{k}_j^{\mathrm{T}}\right) \left(\boldsymbol{v}_j \circ \prod_{t=j+1}^{i} \boldsymbol{f}_t\right)}{\sum_{j=1}^{i} \exp\left(\boldsymbol{q}_i \boldsymbol{k}_j^{\mathrm{T}}\right)} \\
&= \frac{\sum_{j=1}^{i} \exp\left(\boldsymbol{q}_i \boldsymbol{k}_j^{\mathrm{T}}\right) \frac{\boldsymbol{v}_j}{\boldsymbol{\gamma}_j}}{\sum_{j=1}^{i} \exp\left(\boldsymbol{q}_i \boldsymbol{k}_j^{\mathrm{T}}\right)} \circ \boldsymbol{\gamma}_i.
\end{aligned}
\tag{13}
$$

Figure 1 shows how the output $\boldsymbol{o}_3$ is computed: each value vector $\boldsymbol{v}_j$ is first gated element-wise by the cumulative forget gate $\prod_{t=j+1}^{3} \boldsymbol{f}_t$, then weighted by the attention score $\exp(\boldsymbol{q}_3 \boldsymbol{k}_j^{\mathrm{T}})$ and summed. When $j = 3$, the cumulative forget gate $\prod_{t=4}^{3} \boldsymbol{f}_t$ is an empty product equal to a vector of ones, so $\boldsymbol{v}_3$ is not changed.

### 3.1. Fine-grained Forget Gate in *FiX*

The simplest method to generate a fine-grained forget gate is to apply a sigmoid function to a linear projection, i.e., $\boldsymbol{f}_i = \mathrm{sigmoid}\left(\boldsymbol{x}_i \boldsymbol{W}_f + \boldsymbol{b}_f\right)$. However, this approach has two main drawbacks: the linear projection introduces an excessive number of extra parameters, and the sigmoid function has a limited numerical range. To address these issues, we first employ a low-rank projection to reduce the number of parameters. Second, to achieve a wider dynamic range, we adopt an activation function adopted by Mamba (Gu & Dao, 2024). Combining these two improvements, our fine-grained forget gate is formulated as:

$$
\begin{aligned}
\boldsymbol{x}_i^{(f)} &= \mathrm{RMSNorm}\left(\boldsymbol{x}_i \boldsymbol{W}_f^{\downarrow}\right) \boldsymbol{W}_f^{\uparrow} + \boldsymbol{b}_f, \\
f_{i,j} &= \mathrm{sigmoid}^a\left(-x_{i,j}^{(f)}\right),
\end{aligned}
\tag{14}
$$

where $\boldsymbol{W}_f^{\downarrow}$, $\boldsymbol{W}_f^{\uparrow}$, $\boldsymbol{b}_f$ and $\log a$ are learnable parameters. The presence of the negative sign $(-)$ is for the convenience of using the softplus $(\cdot)$ function to calculate $\log \boldsymbol{f}$.

This method works for most layers except for the first layer. This is because it is difficult to learn a projection layer of fine-grained forget gate for the discrete embedding input in the first layer. In practice, the L2 norm of the forget gate projection's gradient in the first layer is not bounded, causing huge unstablity in the training process. Thus, for the first layer we learn an embedding module for the fine-grained forget gate instead of a projection layer. The forget

gate of the first layer is generated by the following equation:

$$\boldsymbol{f}_i = \text{sigmoid}\left(-\boldsymbol{E}_f\left[i_t\right]\right), \tag{15}$$

where $\boldsymbol{E}_f$ is the learnable embedding module for the forget gate and $i_t$ is the token id (an integer) of the token in time step $t$. Note that though this extra embedding introduces more parameters, they cause no computation and can be easily offloaded to CPU memory during inference time.

**RMSNorm configuration.** We highlight that the $\epsilon$ used in the RMSNorm layer followed by the attention output in *FiX* should be much smaller than that in other models, considering that Equation 12 only holds when $\epsilon = 0$. In practice, most open-source SOTA LLMs (Grattafiori et al., 2024; DeepSeek-AI et al., 2025; Team et al., 2025) usually set $\epsilon$ to $10^{-5}$ or $10^{-6}$ in order to prevent division by zero and keep numerical stability. Instead of common settings, we use $\epsilon = 10^{-30}$ to satisfy the requirement of *FiX*. Since $\epsilon$ is much smaller than usual, it causes more numerical instability. As such, we fuse the RMSNorm into the attention implementation to normalize the output of attention using `float32` format to keep higher precision, which is discussed in Section 4.2.

### 3.2. Numerical Stability

On modern GPUs, efficient training relies on Tensor Cores, specialized hardware that exclusively accelerates matrix multiplication. The computation of $\boldsymbol{o}$ in *FiX* can be expressed in two forms. The matrix form is suitable for Tensor Cores but is numerically unstable due to division by very small values. Conversely, the parallel form is numerically stable but cannot be computed using matrix multiplication. In this section, we first analyze the numerical instability of the matrix form and the computational limitations of the parallel form. We then introduce a rescaling trick to derive a formulation that is both numerically stable and can be computed efficiently as a matrix multiplication.

#### 3.2.1. MATRIX FORM

We can rewrite Equation 13 in the matrix form as follows:

$$\boldsymbol{O} = \left(\text{Softmax}\left(\boldsymbol{Q}\boldsymbol{K}^{\text{T}} + \boldsymbol{M}\right)\left(\frac{\boldsymbol{V}}{\boldsymbol{\Gamma}}\right)\right) \circ \boldsymbol{\Gamma}, \tag{16}$$

where $\boldsymbol{Q}, \boldsymbol{K} \in \mathbb{R}^{s \times d_k}$, $\boldsymbol{V}, \boldsymbol{O}, \boldsymbol{\Gamma} \in \mathbb{R}^{s \times d_v}$, $d_k$ is the dimension of key and query, $d_v$ is the dimension of value and output, and $s$ is the sequence length. Each row of these matrices corresponds to a vector at that time step, e.g., the $i$-th row of $\boldsymbol{Q}$ stands for $\boldsymbol{q}_i$. Especially, $\boldsymbol{\Gamma}$ can be computed by a cumulative product of $\boldsymbol{F}$ along the sequence dimension, where the $i$-th row of $\boldsymbol{F}$ is $\boldsymbol{f}_i$. The $\boldsymbol{M} \in \mathbb{R}^{s \times s}$ is the causal mask matrix, whose elements are $0$ in the lower triangle and $-\infty$ in the strict upper triangle.

If we define $\boldsymbol{V}' = \frac{\boldsymbol{V}}{\boldsymbol{\Gamma}}$ and $\boldsymbol{O}' = \frac{\boldsymbol{O}}{\boldsymbol{\Gamma}}$, where the division is element-wise, we can compute $\boldsymbol{O}$ in the following steps:

$$\begin{aligned} \boldsymbol{V}' &= \frac{\boldsymbol{V}}{\boldsymbol{\Gamma}}, \\ \boldsymbol{O}' &= \text{Softmax}\left(\boldsymbol{Q}\boldsymbol{K}^{\text{T}} + \boldsymbol{M}\right)\boldsymbol{V}', \\ \boldsymbol{O} &= \boldsymbol{O}' \circ \boldsymbol{\Gamma}. \end{aligned} \tag{17}$$

The second line of Equation 17 is the standard causal softmax attention. In this matrix form, the entire sequence is processed at once using a single, transformed value matrix $\boldsymbol{V}'$. This allows the use of a single, large matrix multiplication $\boldsymbol{Q}\boldsymbol{K}^{\text{T}}$ followed by another with $\boldsymbol{V}'$, which is highly efficient on GPU tensor cores. However, since $\boldsymbol{\gamma}_i = \prod_{j=1}^i \boldsymbol{f}_j$ and each element in $\boldsymbol{f}_j$ is in the range of $(0, 1)$, the value in $\boldsymbol{\gamma}_i$ can be very small when $i$ is large. Thus, the value in $\boldsymbol{V}'$ might explode, causing numerical instability. This means that materializing $\boldsymbol{V}'$ and then using existing kernel for standard softmax attention is infeasible. Therefore, *FiX* needs a more numerically stable implementation.

#### 3.2.2. FALLBACK TO PARALLEL FORM

Due to the numerical instability in the matrix form, we have to resort to the parallel form as in the first line of Equation 13. Nonetheless, while the parallel form eliminates the need for division, it cannot leverage the full parallel computing power of GPU Tensor Cores. Specifically, for each query vector $\boldsymbol{q}_i$, the corresponding value vectors $\boldsymbol{v}_j$ from previous steps are not the same. Taking $\boldsymbol{v}_1$ as an example, in *FiX*, $\boldsymbol{q}_1$ uses $\boldsymbol{v}_1$ but $\boldsymbol{q}_2$ requires $\boldsymbol{v}_1 \circ \boldsymbol{f}_2$, and so on. Thus, we can not combine all the transformed values into one common $\boldsymbol{V}$ matrix. Instead, the calculation must be done separately for each query position. This results in many independent *matrix-vector* multiplication operations, rather than a single *matrix-matrix* multiplication operation.

#### 3.2.3. RESCALE TRICK

Though it seems that the numerical stability and matrix multiplication are conflict to each other, we can use a rescale trick to balance them.

Suppose $m \leq i$, then

$$\begin{aligned} \boldsymbol{o}_i &= \frac{\left[\sum_{j=1}^m \exp\left(\boldsymbol{q}_i \boldsymbol{k}_j^{\text{T}}\right) \boldsymbol{v}_j \circ \frac{\boldsymbol{\gamma}_m}{\boldsymbol{\gamma}_j}\right]}{\sum_{j=1}^i \exp\left(\boldsymbol{q}_i \boldsymbol{k}_j^{\text{T}}\right)} \circ \frac{\boldsymbol{\gamma}_i}{\boldsymbol{\gamma}_m} \\ &+ \frac{\sum_{j=m+1}^i \exp\left(\boldsymbol{q}_i \boldsymbol{k}_j^{\text{T}}\right) \boldsymbol{v}_j \circ \frac{\boldsymbol{\gamma}_i}{\boldsymbol{\gamma}_j}}{\sum_{j=1}^i \exp\left(\boldsymbol{q}_i \boldsymbol{k}_j^{\text{T}}\right)}. \end{aligned} \tag{18}$$

The above equation avoids division by small numbers. In the first line, the condition $j \leq m \leq i$ ensures that the ratios $\boldsymbol{\gamma}_m/\boldsymbol{\gamma}_j$ and $\boldsymbol{\gamma}_i/\boldsymbol{\gamma}_m$ are less than 1. Furthermore, for a given $m$ and any query $\boldsymbol{q}_i$ where $i \geq m$, the term $\boldsymbol{v}_j \circ \frac{\boldsymbol{\gamma}_m}{\boldsymbol{\gamma}_j}$ is constant. This property allows the summation

in the first line of Equation 18 to be computed as a matrix multiplication.

The FlashAttention (Dao, 2024) algorithm naturally splits $Q$ and $O$ into chunks in its outer loop. For each chunk, we select the largest $m$ such that $m \leq i$ for any index $i$ in that chunk. This approach allows most of the computation to be performed using matrix multiplication while maintaining numerical stability. The details of the efficient implementation for Equation 18 are discussed in Section 4.

### 3.3. Connection to Related Work

**Relations with FoX.** It is evident that VO-FoX is a special case of *FiX* when $\boldsymbol{f}_i$ falls back to a scalar forget gate. Meanwhile, as previously discussed in Section 2.4, the VO-FoX is equivalent to the standard FoX (QK-FoX) under RMSNorm. Thus, *FiX* can be viewed as a generalization of FoX, which fixes the lack of fine-grained forget gate.

**Relations with positional encoding on QK.** Existing QK positional encoding falls mainly into two categories (Zhang et al., 2025a): *additive* and *multiplicative*. Specifically, additive position encoding, such as ALiBi (Press et al., 2022) and FoX (Lin et al., 2025), adds a bias term to the attention logit, i.e., $\boldsymbol{q}_i \boldsymbol{k}_j^{\mathrm{T}} + D_{i,j}$, where $D_{i,j}$ is a position-dependent bias. Multiplicative position encoding, such as RoPE (Su et al., 2024) and PaTH (Yang et al., 2025c), applies a transformation between $\boldsymbol{q}_i$ and $\boldsymbol{k}_j^{\mathrm{T}}$, i.e., $\boldsymbol{q}_i \boldsymbol{T}_{i,j} \boldsymbol{k}_j^{\mathrm{T}}$, where $\boldsymbol{T}_{i,j}$ is a position-dependent transformation matrix. *FiX* is compatible with them as they are all applied to $\boldsymbol{q}$ and $\boldsymbol{k}$ while *FiX* is applied to $\boldsymbol{v}$ and $\boldsymbol{o}$. For example, combined with RoPE, all $\boldsymbol{q}, \boldsymbol{k}, \boldsymbol{v}$, and $\boldsymbol{o}$ receive appropriate position encoding:

$$\boldsymbol{o}_i = \frac{\sum_{j=1}^{i} \exp\left(\boldsymbol{q}_i \boldsymbol{R}_{i-j} \boldsymbol{k}_j^{\mathrm{T}}\right) \left(\boldsymbol{v}_j \circ \prod_{t=j+1}^{i} \boldsymbol{f}_t\right)}{\sum_{j=1}^{i} \exp\left(\boldsymbol{q}_i \boldsymbol{R}_{i-j} \boldsymbol{k}_j^{\mathrm{T}}\right)}, \quad (19)$$

where $\boldsymbol{R}_{i-j}$ is the RoPE rotation for relative position $i - j$.

## 4. Efficient and Stable Implementation

In this part, we first introduce our Flash Fine-grained Attention implementation in Section 4.1, and then discuss how we fuse the RMSNorm into the attention and linear kernels for numerical stability in Section 4.2. Finally, we explain how to reduce the memory consumption of the extra F cache during inference in Section 4.3.

### 4.1. Flash Fine-grained Attention

Standard softmax attention, while highly parallelizable, generates intermediate matrices of size $s \times s$, where $s$ is the sequence length. The $O(s^2)$ memory requirement for these matrices creates a bottleneck in memory access and makes long context training impractical. To overcome this, FlashAttention (Dao, 2024) avoids materializing the full

attention matrix in High-Bandwidth Memory (HBM) by computing the attention output using online softmax algorithm on SRAM. This a memory-efficient approach that has become the standard of attention implementations.

However, as discussed in Section 3.2, we cannot directly use the standard FlashAttention due to numerical stability issues. We therefore propose *Flash Fine-grained Attention*, an adaptation that maintains efficiency while ensuring stability.

Our method follows the tiling strategy of FlashAttention. For simplicity, we assume the query $Q$, key $K$, value $V$, and output $O$ matrices are all partitioned into $n$ blocks of size $B$. An outer loop processes blocks of $Q$ and $O$ in parallel across different thread blocks on the GPU. Within each thread block, an inner loop iterates over the blocks of $K$ and $V$. For a given query block $Q_i$, the first time step in $Q_i$ is $(i-1) \times B + 1$, and we set it to the $m$ in Equation 18. Then, when iterating through key/value blocks $K_j, V_j$, we distinguish two cases for attention computation:

- If $j < i$, all time steps in $V_j$ are smaller than $m$. The computation for this block corresponds to the first term in Equation 18 with standard matrix multiplication.
- If $j = i$, the block $V_j$ contains time steps larger than $m$. Thus, the computation corresponds to the second term in Equation 18, where we compute each $\boldsymbol{o}_i$ individually.

The number of blocks where $j < i$ scales quadratically ($O(s^2)$) with the sequence length, whereas the number of diagonal blocks where $j = i$ scales linearly ($O(s)$). Consequently, the vast majority of the computation can be calculated as matrix multiplication and accelerated using highly optimized GPU Tensor Cores. We implement Flash Fine-grained Attention for both forward and backward passes, with detailed algorithms provided in Appendix E.

### 4.2. Fused RMSNorm and Linear Layer

The Root Mean Square Layer Normalization (RMSNorm) is defined as follows. Given an input vector $\boldsymbol{x} \in \mathbb{R}^{d_v}$ and a weight $\boldsymbol{w} \in \mathbb{R}^{d_v}$, the output $\boldsymbol{y} \in \mathbb{R}^{d_v}$ is computed by:

$$s = \sqrt{\frac{1}{d_v} \sum_{i=1}^{d_v} x_i^2 + \epsilon}, \quad (20)$$
$$\boldsymbol{y} = \frac{\boldsymbol{x}}{s} \circ \boldsymbol{w}.$$

The scaling factor $s$ is in the denominator, which can lead to numerical instability if $s$ is close to zero. A small constant $\epsilon$, typically $10^{-5}$ or $10^{-6}$, is added to prevent this instability.

In our work, we set $\epsilon$ to a much smaller value of $10^{-30}$, which increases the risk of instability. During training, intermediate results are often downcast to `bfloat16` format before being written to HBM from SRAM to conserve mem-

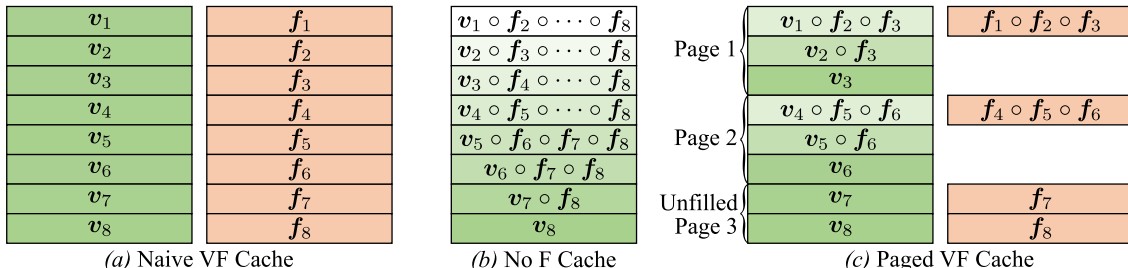

*(a)* Naive VF Cache      *(b)* No F Cache      *(c)* Paged VF Cache

*Figure 2.* Comparison of cache strategies for managing the forget gate (F) and value (V) cache.

ory bandwidth. With our small $\epsilon$, the reduced precision of `bfloat16` is insufficient to preserve numerical stability during the division, and thus this computation must be performed in the `float32` precision format.

Since our attention kernel holds the output in on-chip SRAM using `float32` format before the output is finally written back to HBM, we can fuse the RMSNorm forward pass within the Flash Fine-grained Attention kernel. This allows the normalization to be applied in high precision, avoiding the HBM round-trip and associated precision loss. The similar challenge also arises in the backward pass, because the gradient with respect to $x$, $\frac{\partial \mathcal{L}}{\partial x}$, also involves a division by $s$, as shown in the following formula:

$$
\begin{aligned}
c &= \sum_{i=1}^{d_v} y_i \frac{\partial \mathcal{L}}{\partial y_i}, \\
\frac{\partial \mathcal{L}}{\partial x} &= \frac{1}{s} \left( \frac{\partial \mathcal{L}}{\partial y} \circ w - c \frac{y}{w} \right).
\end{aligned}
\tag{21}
$$

Note that the gradient $\frac{\partial \mathcal{L}}{\partial y}$ is also read from HBM in `bfloat16` format. Therefore, the division by $s$ still causes numerical instability. As the $\frac{\partial \mathcal{L}}{\partial y}$ is the output of the backward pass of the subsequent linear layer, to conduct the division in high precision, we fuse the RMSNorm's backward pass within the linear layer's backward pass to perform the calculation using the on-chip gradients in `float32` format.

Combining these forward and backward requirements, we fuse the attention, RMSNorm, and the subsequent linear layer into a single *FiX*-RMSNorm-Linear operator. This single fused operator ensures the numerical stability throughout the training process. Details of the implementation for the forward and backward passes are provided in Algorithm 3 and Algorithm 4 in Appendix E, respectively.

### 4.3. Paged VF Cache for Inference

Standard auto-regressive inference with attention requires a KV cache to store past keys and values. By introducing a fine-grained forgetting gate, *FiX* additionally requires a element-wise forget gate, which necessitates an *F cache*. However, we find that we can mitigate this extra storage with minimal computation overheads by absorbing the F

cache into V cache with paged management. Note that as *FiX* makes no changes to the key structures, we only focus on the V and F caches in the following discussion.

As shown in Figure 2a, the naive approach is to store the V and F caches separately. However, the forget gate $f$ must be stored in `float32` for numerical stability, while keys and values are typically stored in `bfloat16`. Consequently, this naive F cache doubles the memory requirement by requiring an extra overhead equal to the standard KV cache. An alternative, as depicted in Figure 2b, absorbs the cumulative forget gates directly into the V cache at each step and eliminate the additional F Cache. However, this approach has two major limitations: (1) it doubles the memory bandwidth required for the V Cache, as every $v$ must be read and re-written at each decoding step to incorporate a new $f$; (2) the repeated conversions from `bfloat16` to `float32` for multiplication and back to `bfloat16` for storage cause cumulative quantization errors that degrade value precision.

To balance the memory footprint and numerical precision, we propose *paged VF cache*. Specifically, we manage the V and F caches in fixed-size pages and absorb F cache into V cache within each page. For example, as illustrated in Figure 2c, there are 8 tokens and we set the page size $p$ to 3. For tokens not filling a complete page (namely token 7 and 8), their generated V and F values are stored in a new, unfilled page separately in their own caches. Once a page is completely filled (e.g., page 1 and 2), the cumulative forget gates within that page are absorbed into the values. Absorbed V caches in complete pages are frozen. Besides, we only store the product of the forget gates as the F cache, e.g., $f_1 \circ f_2 \circ f_3$. For subsequent steps, this product is used to scale the values in V cache before that page. For example, $f_4 \circ f_5 \circ f_6$ is used to scale the V caches in page 1.

This paged VF cache offers two benefits: (1) it brings minimal memory overhead to store the products of F values in `float32`, which is roughly $1/p$ of the standard KV cache; (2) it also avoids updating the entire V cache at every step and only requires to update the $v$ vector in a page every $p$ decoded tokens. In practice, we find that setting $p$ to 16 achieves a good balance between computation and memory. Furthermore, this paged management can be seamlessly in-

tegrated into popular inference engines like vLLM (Kwon et al., 2023) and SGLang (Zheng et al., 2024b), since they already use PagedAttention (Kwon et al., 2023). We implement this cache update mechanism, and the algorithm is detailed in Appendix D.

# 5. Experiments

We experiment with *FiX* and compare it against two important position encoding baselines: RoPE (Su et al., 2024) and FoX (Lin et al., 2025). Because *FiX* and FoX can both work with RoPE, we also experiment with *FiX*-RoPE and FoX-RoPE. We train all the models on a subset of FineWeb-Edu (Lozhkov et al., 2024), a collection of educational text filtered from FineWeb. Detailed architectural specifications are provided in Appendix B.

## 5.1. Training Setup

All models were trained using the AdamW (Loshchilov & Hutter, 2019) optimizer with $\beta = (0.9, 0.95)$ and a cosine learning rate schedule. We employed a warmup period of 512M tokens, where the learning rate increased linearly from 0 to a peak of $1 \times 10^{-3}$ before decaying to a final value of $1 \times 10^{-4}$. We applied a weight decay of 0.1, a global gradient clipping of 1.0, and a batch size of 512K tokens. Models with approximately 760M parameters are trained on 48B tokens (92160 steps), while 340M models are trained on 10B tokens (20480 steps). Most parameters were initialized with a standard deviation of 0.02, with the exception of specific forget-gate related parameters.

The initialization of $\boldsymbol{b}_f$ and $\log a$ in Equation 14 follows Mamba (Gu & Dao, 2024), GDN (Yang et al., 2025b), and KDA (Zhang et al., 2025b). The parameters are initialized as the following equation:

$$
\begin{aligned}
A &\sim \text{Uniform}\left(a_{\min}, a_{\max}\right), \\
D_i &\sim \text{Uniform}\left(\log \Delta_{\min}, \log \Delta_{\max}\right), \\
\log a &= \log A \quad, \\
\Delta_i &= \max\left(\exp\left(D_i\right), \Delta_{\text{floor}}\right), \\
b_{f,i} &= \text{softplus}^{-1}\left(\Delta_i\right),
\end{aligned}
\tag{22}
$$

where $\text{softplus}(x) = \log(1 + e^x)$.

In this paper, we set $a_{\min} = 0$, $a_{\max} = 16$, $\Delta_{\min} = 10^{-3}$, $\Delta_{\max} = 10^{-1}$, and $\Delta_{\text{floor}} = 10^{-4}$.

## 5.2. Training Loss

We first compare the training loss of different models on identical training data. Due to stochastic fluctuations across batches, the raw loss curves can be noisy and obscure the subtle yet consistent differences between models. To provide a clearer comparison, we report the loss difference of each model relative to *FiX*.

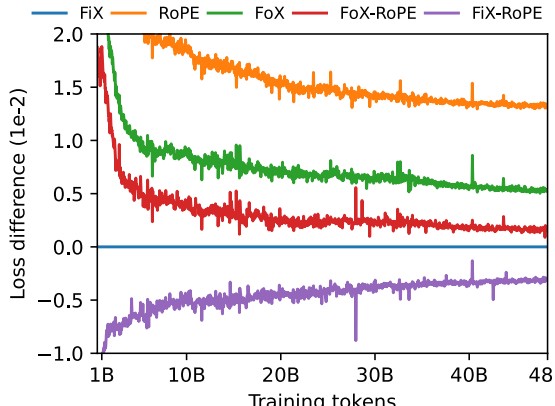

*Figure 3.* Training loss difference relative to *FiX*, calculated as $\mathcal{L}_{\text{model}} - \mathcal{L}_{\text{FiX}}$. Models with positive values (above the zero line) are outperformed by *FiX*.

Figure 3 plots these differences, calculated as $\mathcal{L}_{\text{model}} - \mathcal{L}_{\text{FiX}}$. A positive value (above the zero line) indicates that the model's loss is higher than *FiX*'s, while a negative value signifies better performance than *FiX*. As shown in the figure, RoPE, FoX, and their combination FoX-RoPE all exhibit consistently positive loss differences, indicating that they underperform *FiX* throughout the whole training. In contrast, *FiX*-RoPE the only model that consistently remains in the negative region, achieving a lower loss than standalone *FiX*. These results suggests two key insights: (1) *FiX* is the most effective standalone position encoding method among the ones tested; (2) *FiX* can be effectively combined with RoPE to achieve further performance gains, highlighting its compatibility and synergistic potential.

## 5.3. Common Sense Benchmarks

We evaluate the models on a range of zero-shot common sense reasoning tasks. Specifically, we report the model perplexity on Wikitext (Merity et al., 2016) and LAMBADA (Kazemi et al., 2023), as well as accuracy on LAMBADA, PIQA (Bisk et al., 2020), HellaSwag (Zellers et al., 2019), WinoGrande (Sakaguchi et al., 2020), ARC-easy, ARC-challenge (Clark et al., 2018), SIQA (Sap et al., 2019), and BoolQ (Clark et al., 2019).

The results in Table 1 highlight the strong and consistent performance of *FiX*. Among all models, *FiX* and *FiX*-RoPE achieve the highest average and geometric mean accuracies, demonstrating their superior overall capability. Notably, standalone *FiX* achieves the best average accuracy (55.24) and geometric mean (53.79), yielding the top or second-best result in 7 out of the 10 reported metrics. This consistent performance gain across the diverse set of benchmarks underscores its reliability and robust generalization ability. We notice that *FiX* achieves slightly better results than *FiX*-RoPE despite that *FiX*-RoPE has lower training loss. This is

*Table 1.* Zero-shot evaluation on common sense reasoning benchmarks and language modeling perplexity. We report perplexity (ppl) on Wikitext (Wiki.) and LAMBADA (LMB.), and accuracy (acc/acc_n) for eight downstream tasks. The average (Avg.) and geometric mean (Geo Mean.) are calculated over the accuracies of these eight tasks. Best results are in **bold**, and second-best are underlined.

| Model | Wiki. ppl ↓ | LMB. ppl ↓ | LMB. acc ↑ | PIQA acc ↑ | Hella. acc_n ↑ | Wino. acc ↑ | ARC-e acc ↑ | ARC-c acc_n ↑ | SIQA acc ↑ | BoolQ acc ↑ | Avg. ↑ | Geo Mean. ↑ |
|---|---|---|---|---|---|---|---|---|---|---|---|---|
| RoPE | 18.21 | 13.39 | 46.90 | 71.33 | 55.12 | 59.35 | 67.63 | 36.26 | **41.30** | 60.64 | 54.82 | 53.51 |
| FoX | 17.88 | 13.35 | 45.82 | 71.55 | 55.83 | 59.12 | 68.81 | 34.98 | 40.69 | 58.56 | 54.42 | 52.98 |
| FoX-RoPE | **17.66** | 13.14 | 46.23 | 72.20 | 56.05 | **60.54** | **70.58** | **36.52** | 39.97 | 55.26 | 54.67 | 53.23 |
| *FiX* | 17.72 | 12.69 | 47.78 | **72.58** | **56.32** | 60.06 | 68.35 | 35.32 | 40.63 | **60.86** | **55.24** | **53.79** |
| *FiX*-RoPE | 17.85 | **12.32** | **48.55** | 70.89 | 56.25 | 59.35 | 69.23 | 36.09 | 40.58 | 57.80 | 54.85 | 53.52 |

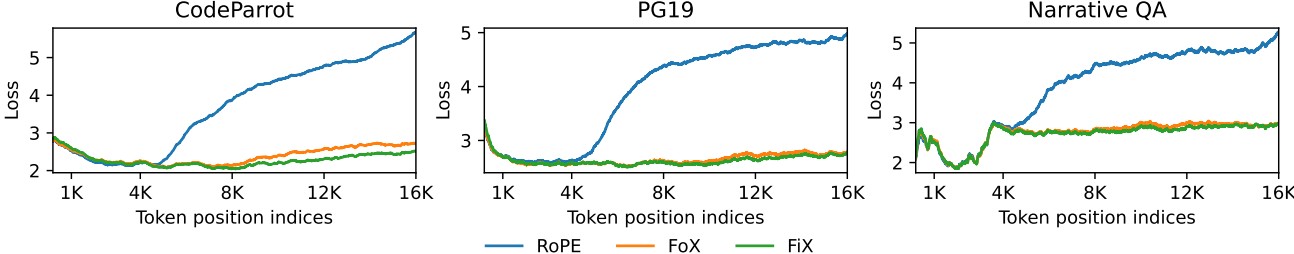

*Figure 4.* Length extrapolation performance on CodeParrot, PG-19, and NarrativeQA. Models were trained with a context length of 4096 and evaluated on sequences up to 16,384. The metric is the average loss at each token position.

because the pretraining loss of a model does not necessarily reflect its performance on certain downstream tasks.

## 5.4. Long Context Tasks

**Length extrapolation.** In Figure 4, we evaluate the models' ability to handle contexts longer than those seen during training, without any fine-tuning. We compare *FiX* to RoPE, FoX on three long-context datasets: PG-19 (books) (Rae et al., 2020), CodeParrot (code), and NarrativeQA (dialogue) (Kočiský et al., 2018). Following FoX (Lin et al., 2025), we use per-token loss to sketch the ability of length extrapolation. We train all models with a context length of 4096 and tested them on sequences up to 16,384, measuring the average loss at each token position. The results show that both FoX and *FiX* maintain stable and low loss up to 16,384 tokens. Notably, *FiX* consistently performs slightly better than FoX across all three datasets, demonstrating its superior long-context modeling capabilities. In constrast, RoPE's performance collapses sharply when the context length exceeds the 4096-token training limit, suggesting its inability to extrapolate. As this catastrophic failure would dominate the results, we do not combine RoPE with *FiX* and FoX in this experiment.

**BABILong.** To further assess the long-context reasoning ability, we evaluate all models on the BABILong (Kuratov et al., 2024) benchmark, which tests a model's ability to reason about facts distributed across extremely long documents. Figure 5 reports the average accuracy across BABILong's five question-answering (QA) tasks as context length increases.

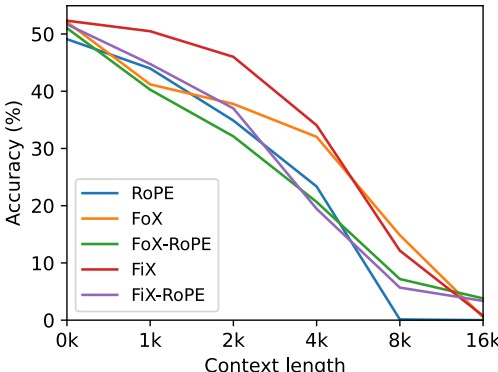

*Figure 5.* Average accuracy on the BABILong benchmark across varying context lengths. *FiX* demonstrates superior performance, especially in the 1k-4k context range.

The results show that *FiX* consistently demonstrates a good model accuracy, significantly outperforming other baselines, particularly in the 1k-4k context length range. We also decompose the results on 5 different types of QA questions in BABILong, detailed in Appendix C.1. We observe that *FiX* outperforms baselines on most of the types, further validating its effectiveness.

## 5.5. Ablation Study

We conduct an ablation study to justify three key design choices in *FiX*: (1) setting $\epsilon = 10^{-30}$ in the output RM-SNorm, (2) using a forget gate embedding (Equation 15) in the first layer, and (3) employing a Mamba-style acti-

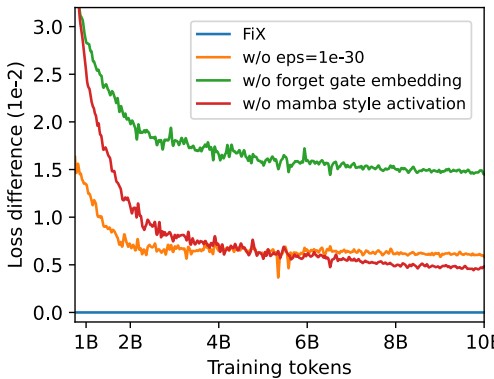

*Figure 6.* Ablation study of *FiX*'s design choices. The plot shows the training loss difference relative to the full *FiX* model ($\mathcal{L}_{\text{ablated}} - \mathcal{L}_{\text{FiX}}$). All ablated variants perform worse, confirming the importance of each component.

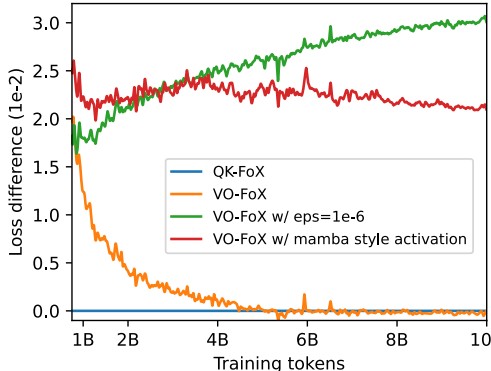

*Figure 7.* Study of the equivalence of VO-FoX and QK-FoX. The plot shows the training loss difference relative to the FoX model ($\mathcal{L}_{\text{ablated}} - \mathcal{L}_{\text{FoX}}$).

vation (Equation 14) for the forget gate. We create the three model variants by reverting each of these designs from our proposed *FiX* architecture, specifically: (1) replacing $\epsilon = 10^{-30}$ with a standard value of $\epsilon = 10^{-6}$, (2) using a low-rank projection for the forget gate in the first layer instead of an embedding, (3) using a simple sigmoid activation for the forget gate.

We report the training loss difference of these ablated models relative to the complete *FiX* model in Figure 6. The results show that all three variants have positive loss differences, indicating that their performance is worse than that of the full *FiX* model. This confirms that all three design choices are crucial to achieve the optimal performance.

We further support the necessity of these components through theoretical analysis. In particular, the small $\epsilon = 10^{-30}$ is essential for maintaining the equivalence between VO-FoX and QK-FoX. The forget gate embedding in the first layer is required to stabilize training, as a standard projection layer at this stage suffers from an exploding gradient norm. Finally, while the Mamba-style activation enhances performance, it is only effective for fine-grained gates, not scalar ones. This suggests that the primary performance gain stems from our novel fine-grained forget gate.

**Equivalence of VO-FoX and QK-FoX.** Figure 7 shows the loss difference compared to FoX. When $\epsilon$ is set to $10^{-30}$, the loss difference of VO-FoX and FoX converges to 0. This proves the equivalence of VO-FoX and QK-FoX when $\epsilon$ is close to 0. In contrast, when $\epsilon$ is set to $10^{-6}$, the VO-FoX is outperformed by FoX, which again shows the importance of setting $\epsilon$ to $10^{-30}$.

**The effect of Mamba-style activation function.** In Figure 7, we keep $\epsilon = 10^{-30}$ in VO-FoX and change the activation function of the forget gate from simple sigmoid function to Mamba-style activation function. Different with the result presented in Figure 6, the Mamba-style activation

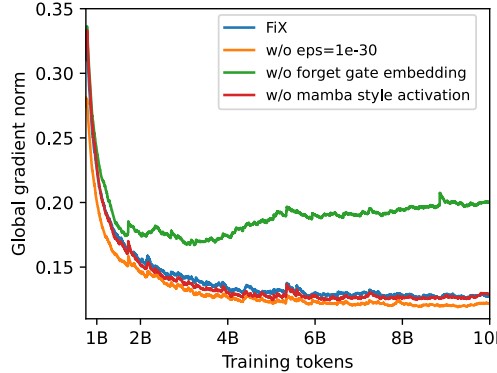

*Figure 8.* Ablation study of *FiX*'s design choices. The plot shows the global gradient norm against the training steps.

function performs worse than sigmoid function when only using a scalar forget gate. This verifies that the performance gain of *FiX* originates from the our fine-grained forget gate rather than Mamba-style activation function, even though it is necessary for *FiX*.

## 6. Conclusion

This paper introduces *FiX*, which incorporates a fine-grained forget gate into softmax attention by applying it to the $\boldsymbol{v}$ and $\boldsymbol{o}$ vectors. Notably, *FiX* is the first positional encoding on V-O vectors as far as we understand. For efficient and numerically stable training and inference, we implement fused Flash Fine-grained Attention and a paged VF cache. We compare *FiX* against RoPE and FoX on various tasks and experimental results demonstrate that *FiX* outperforms these baselines both in terms of training loss and model accuracy. We also show that *FiX* can be seamlessly combined with RoPE to produce further improvements.

## Acknowledgments

This work is partially supported by the National Natural Science Foundation of China (Grant Nos. 62422206, 62532007, 62532001) and the Hong Kong Research Grants Council under the Theme-based Research Scheme (project no. T41-517/25-N). This work is also a research gift from AlayaDB.AI Inc.

## Impact Statement

This work is focused on advancing the architecture of neural networks. By proposing a method to create more powerful transformers, our research contributes to the general progress in artificial intelligence. While this work is primarily aimed to accelerate positive applications in science, education, and accessibility, we acknowledge that more capable models also inherit the risks associated with large language models, such as the potential generation of sophisticated misinformation. We advocate for the continued development of safety protocols and responsible deployment practices in parallel with such performance improvements.

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

# A. Related Work

Existing positional encodings in softmax attention can be broadly classified into two categories: additive and multiplicative, as detailed in Section 3.3. Multiplicative methods construct a matrix $\boldsymbol{T}_{ij}$ to modulate the query-key interaction, forming $\boldsymbol{q}_i \boldsymbol{T}_{ij} \boldsymbol{k}_j^{\mathrm{T}}$. Early approaches such as RoPE (Su et al., 2024) and APE (Kogkalidis et al., 2024) employ data-independent constructions for $\boldsymbol{T}_{ij}$. In contrast, more recent work like PaTH (Yang et al., 2025c) introduces data-dependent transformations using Householder-like matrices. Additive methods, on the other hand, directly modify the attention logits. Early works in this category, including ALiBi (Press et al., 2022), Kerple (Chi et al., 2022), and FIRE (Li et al., 2024), add a data-independent bias $b_{ij} = f(i,j)$ to the logits $\boldsymbol{q}_i \boldsymbol{k}_j^{\mathrm{T}}$. Subsequent methods like DAPE (Zheng et al., 2024a), FoX (Lin et al., 2025), CABLE (Veisi et al., 2025), and Selective Attention (Leviathan et al., 2025) incorporate data-dependent biases of the form $b_{ij} = f(i, j, \boldsymbol{x}_j, \ldots, \boldsymbol{x}_i)$. GRAPE (Zhang et al., 2025a) provides a unifying perspective on these two categories through the lens of group theory. Beyond these, alternative mechanisms such as Stick-breaking Attention (Tan et al., 2025) (a causal variant of Geometric Attention (Csordás et al., 2022)) replace the softmax function with a stick-breaking process to generate attention scores.

All these methods focus on the attention logits or scores, effectively making them positional encoding for the query-key (QK) component. While some approaches, such as RoPE, can be mathematically adapted for the value-output (VO) computation as demonstrated in VO-RoPE (Su, 2025), their performance in this context is inferior to the original QK application. In fact, combining VO-RoPE with QK-RoPE often yields worse results than using QK-RoPE alone. In contrast, *FiX* integrate positional information into the VO computation by introducing a fine-grained forget gate, which makes *FiX* the first state-of-the-art positional encoding for $\boldsymbol{v}$ and $\boldsymbol{o}$ vectors.

# B. Architectures

All attention variants were tested under a modern and consistent transformer architecture (equipped with SwiGLU (Shazeer, 2020), QK-Norm (Dehghani et al., 2023), O-Norm, and Short-Conv (Gu & Dao, 2024)). This ensures that any performance differences can be directly attributed to the attention mechanisms. The models used in the primary experiments have approximately 760M parameters, whereas those used for ablation studies have approximately 340M parameters. The 760M-parameter model has 36 hidden layers, 10 heads, and a SwiGLU intermediate size of 3840. The 340M-parameter model has 24 hidden layers, 8 heads, and a SwiGLU intermediate size of 3072. The bottleneck dimension of the low-rank forget gate projection in *FiX* is 128. Thus, the low-rank projection only introduces approximately 8-10M parameters.

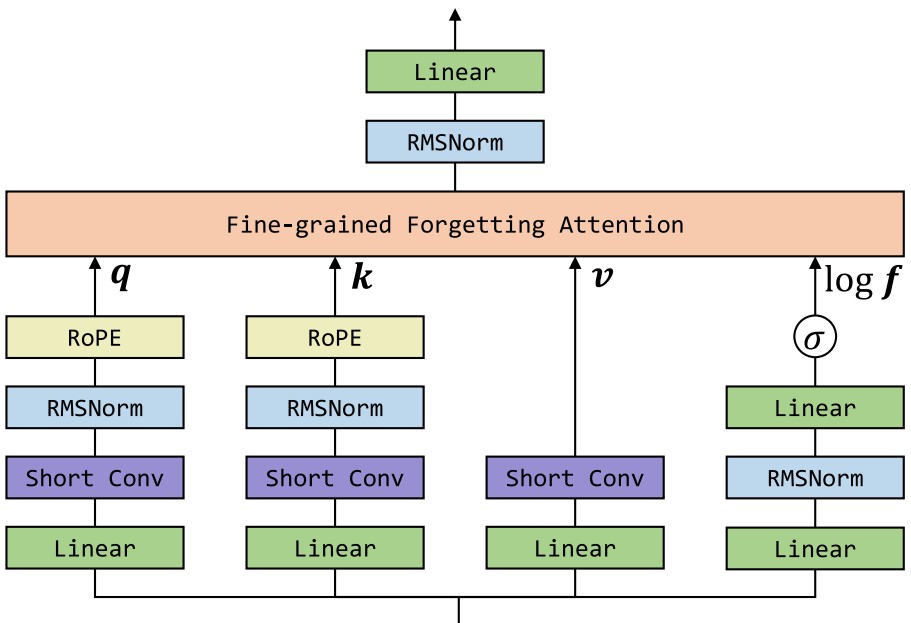

*Figure 9.* The architecture of the attention layer in *FiX* combined with RoPE

Figure 9 illustrates the model architecture of the attention layer when our proposed *FiX* is used in conjunction with Rotary

Position Embeddings (RoPE). The activation function, denoted by $\sigma$ in the diagram, is described in Equation 14. This architecture is adapted for the different models in our experiments.

For models that do not employ RoPE, the corresponding module is simply omitted. When implementing the FoX model, the "Fine-grained Forgetting Attention" is replaced by "Forgetting Attention", wherein $\log f$ is a scalar generated by a single linear layer with a simple sigmoid activation. For the model that only uses RoPE, the attention layer reverts to a standard causal softmax attention mechanism.

Apart from these specified modifications, all models evaluated in our experiments share this common architectural framework to ensure a fair comparison.

## C. Experiments Continued

### C.1. BABILong

Figure 10 presents the accuracy of different models across individual QA tasks in BABILong.

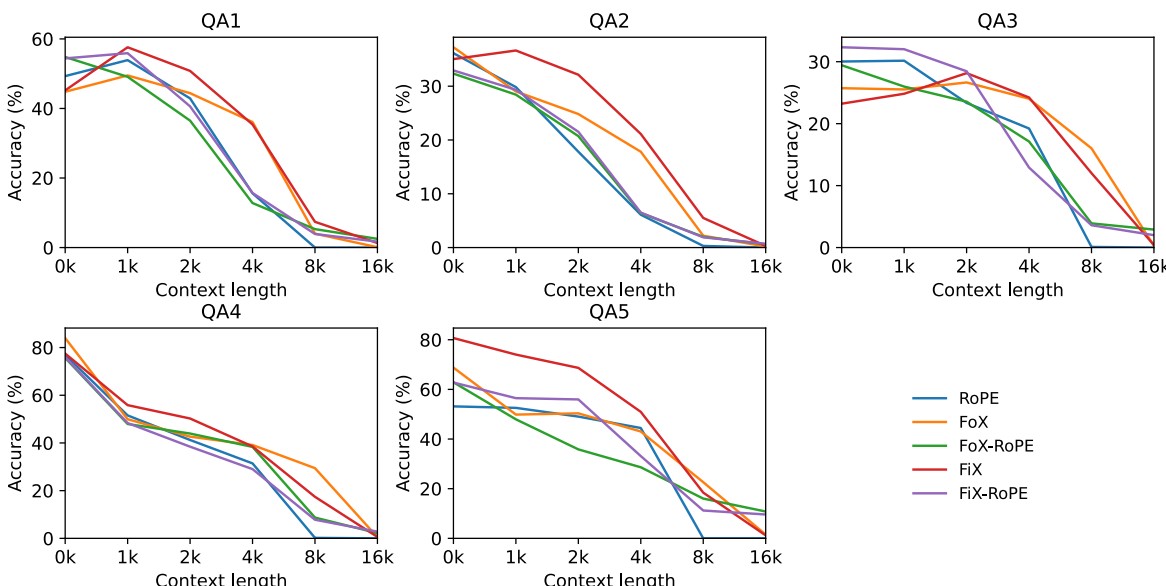

*Figure 10.* Accuracy on individual BABILong tasks across varying context lengths.

**Gradient Norm.** Figure 8 presents the global gradient norms of different models vs. the training steps. The global gradient norms is defined as $\sqrt{\sum_{i=1}^{n} x_i^2}$, where $n$ is the total number of parameters of a model, and $x_i$ is the $i$-th parameter. It shows that if we do not adopt the forget gate embedding, the gradient norm will be unbounded, because it is hard to learn a meaningful projection layer from the discrete token embeddings.

## D. Update Algorithm of Paged VF Cache

We present the algorithm of VF cache update during the prefilling stage in Algorithm 1 and decodeing stage in Algorithm 2.

---

**Algorithm 1** VF Cache Update of *FiX* during Prefilling Stage

---

1: **Global State:**
2: $K_{\text{cache}}, V_{\text{cache}}$: Key/Value Paged Cache tensors.
3: $S_{\text{cache}}$: Cache for sum of $\log f$ for each full page.
4: $F_{\text{current}}$: Cache for $\log F$ values of the last, unfilled page.
5: $M[s, p]$: Mapping from sample $s$, page index $p$ to a global page ID.
6: $N[s]$: Number of pages for sample $s$.
7: $L[s]$: Current length for sample $s$.

8: $i$: The first available page index.

9: $B$: Page size as a constant.

10: **Input:** Batch of sequences $\boldsymbol{K}, \boldsymbol{V}, \log \boldsymbol{F}$ and number of samples $n$.

11: **for** $s = 0$ **to** $n - 1$ **do**

12:      Get the input of sample $s$: $\boldsymbol{K}^{(s)}, \boldsymbol{V}^{(s)}, \log \boldsymbol{F}^{(s)}$ from $\boldsymbol{K}, \boldsymbol{V}, \log \boldsymbol{F}$

13:      Get the length $l_s$ of sample $s$.

14:      $\boldsymbol{L}[s] \leftarrow l_s$.

15:      $p_s \leftarrow \lceil \frac{l_s}{B} \rceil$ is the number of pages for sample $s$.

16:      $\boldsymbol{N}[s] \leftarrow p_s$.

17:      **for** $p = 0$ **to** $p_s - 1$ **do**

18:          Get the input slice for page $p$: $\boldsymbol{K}_p^{(s)}, \boldsymbol{V}_p^{(s)}, \log \boldsymbol{F}_p^{(s)}$ from $\boldsymbol{K}^{(s)}, \boldsymbol{V}^{(s)}, \log \boldsymbol{F}^{(s)}$.

19:          $i_p \leftarrow i$ is the page index for $p$.

20:          $\boldsymbol{M}[s, p] \leftarrow i_p$.

21:          $i \leftarrow i + 1$.

22:          $\boldsymbol{K}_{\text{cache}}[i_p] \leftarrow \boldsymbol{K}_p^{(s)}$.

23:          **if** $p < p_s - 1$ **then**

24:             $\log \boldsymbol{\Gamma}_p^{(s)} \leftarrow \text{CumulativeSum}\left(\log \boldsymbol{F}_p^{(s)}\right)$.

25:             $\boldsymbol{S}_p \leftarrow \log \boldsymbol{\Gamma}_p^{(s)}[-1]$.

26:             $\boldsymbol{V}_{\text{cache}}[i_p] \leftarrow \boldsymbol{V}_p^{(s)} \odot \exp(\boldsymbol{S}_p - \log \boldsymbol{\Gamma}_p^{(s)})$.

27:             $\boldsymbol{S}_{\text{cache}}[i_p] \leftarrow \boldsymbol{S}_p$.

28:          **else**

29:             $\boldsymbol{V}_{\text{cache}}[i_p] \leftarrow \boldsymbol{V}_p^{(s)}$.

30:             $\boldsymbol{F}_{\text{current}}[s] \leftarrow \log \boldsymbol{F}_p^{(s)}$.

31:          **end if**

32:      **end for**

33: **end for**

---

**Algorithm 2** VF Cache Update of *FiX* during Decoding Stage

1: **Global State:**

2: $\boldsymbol{K}_{\text{cache}}, \boldsymbol{V}_{\text{cache}}$: Key/Value Paged Cache tensors.

3: $\boldsymbol{S}_{\text{cache}}$: Cache for sum of $\log \boldsymbol{F}$ for each full page.

4: $\boldsymbol{F}_{\text{current}}$: Cache for $\log \boldsymbol{F}$ values of the last, unfilled page.

5: $\boldsymbol{M}[s, p]$: Mapping from sample $s$, page index $p$ to a global page ID.

6: $\boldsymbol{N}[s]$: Number of pages for sample $s$.

7: $\boldsymbol{L}[s]$: Current length for sample $s$.

8: $i$: The first available page index.

9: $B$: Page size as a constant.

10: **Input:** New tokens $\boldsymbol{k}_n, \boldsymbol{v}_n, \log \boldsymbol{f}_n$ and number of samples $n$.

11: **for** $s = 0$ **to** $n - 1$ **do**

12:      Get the current length $l_s \leftarrow \boldsymbol{L}[s]$.

13:      **if** $l_s > 0$ and $l_s \equiv 0 \pmod{B}$ **then**

14:          $p_{\text{prev}} \leftarrow l_s / B - 1$.

15:          $i_p \leftarrow \boldsymbol{M}[s, p_{\text{prev}}]$.

16:          $\log \boldsymbol{F}_{\text{page}} \leftarrow \boldsymbol{F}_{\text{current}}[s]$.

17:          $\boldsymbol{V}_{\text{page}} \leftarrow \boldsymbol{V}_{\text{cache}}[i_p]$.

18:          $\log \boldsymbol{\Gamma}_{\text{page}} \leftarrow \text{CumulativeSum}(\log \boldsymbol{F}_{\text{page}})$.

19:          $\boldsymbol{S}_{\text{page}} \leftarrow \log \boldsymbol{\Gamma}_{\text{page}}[-1]$.

20:          $\boldsymbol{V}_{\text{cache}}[i_p] \leftarrow \boldsymbol{V}_{\text{page}} \odot \exp(\boldsymbol{S}_{\text{page}} - \log \boldsymbol{\Gamma}_{\text{page}})$.

21:          $\boldsymbol{S}_{\text{cache}}[i_p] \leftarrow \boldsymbol{S}_{\text{page}}$.

22:      **end if**

23:      Get the new token for sample $s$: $\boldsymbol{k}_n^{(s)}, \boldsymbol{v}_n^{(s)}, \log \boldsymbol{f}_n^{(s)}$ from $\boldsymbol{k}_n, \boldsymbol{v}_n, \log \boldsymbol{f}_n$.

24:    $p \leftarrow \lfloor l_s / B \rfloor$ is the page index for the new token.
25:    $j \leftarrow l_s \mod B$ is the index within the page.
26:    **if** $j = 0$ **then**
27:        $i_p \leftarrow i$.
28:        $\boldsymbol{M}[s, p] \leftarrow i_p$.
29:        $\boldsymbol{N}[s] \leftarrow p + 1$.
30:        $i \leftarrow i + 1$.
31:    **else**
32:        $i_p \leftarrow \boldsymbol{M}[s, p]$.
33:    **end if**
34:    $\boldsymbol{K}_{\text{cache}}[i_p, j] \leftarrow \boldsymbol{k}_n^{(s)}$.
35:    $\boldsymbol{V}_{\text{cache}}[i_p, j] \leftarrow \boldsymbol{v}_n^{(s)}$.
36:    $\boldsymbol{F}_{\text{current}}[s, j] \leftarrow \log \boldsymbol{f}_n^{(s)}$.
37:    $\boldsymbol{L}[s] \leftarrow l_s + 1$.
38: **end for**

# E. Flash Fine-grained Attention

We present the implementation of Flash Fine-grained Attention in this section using runnable pseudo-code. The whole implementation includes 4 kernels and a function that integrates these kernels.

Algorithm 3 presents the forward pass of Flash Fine-grained Attention.

---

**Algorithm 3** The Forward Pass of Flash Fine-grained Attention

---

```
def flash_fine_grained_attention_forward(
    q: Tensor,               # [h, s, d_k], bfloat16
    k: Tensor,               # [h, s, d_k], bfloat16
    v: Tensor,               # [h, s, d_v], bfloat16
    log_g: Tensor,           # [h, s, d_v],  float32
    rmsnorm_weight: Tensor, #      [d_v], bfloat16
    sm_scale: float,
    eps: float,
    BM: int = 64,
    BN: int = 64,
):
    # dimensions
    h, s, d_k = q.shape
    d_v = v.shape[-1]

    # final output
    m   = empty(h, s,      dtype= float32)
    l   = empty(h, s,      dtype= float32)
    rms = empty(h, s,      dtype= float32)
    o   = empty(h, s, d_v, dtype=bfloat16)

    # outer loop, executed in parallel across different thread blocks
    for i in range((s + BM - 1) // BM):
        # load q and log_g for block 'i' from HBM, and calculate the indices of q and o
        real_bm = BM if (i + 1) * BM < s else s - i * BM
        indices_m        = i * BM + arange(0, BM)

        bq        = zeros(h, BM, d_k, dtype=bfloat16)
        b_log_g_m = zeros(h, BM, d_v, dtype= float32)

        bq        [:,:real_bm].copy_(q    [:,i * BM: i * BM + real_bm])
        b_log_g_m[:,:real_bm].copy_(log_g[:,i * BM: i * BM + real_bm])

        # variables used in online softmax:
        # row-wise maximum, denominator of softmax, output
        bm = full ([h, BM]    , -inf, dtype=float32)
        bl = zeros([h, BM]    ,       dtype=float32)
        bo = zeros([h, BM, d_v],      dtype=float32)
```

```
# inner loop
# 'i * BM' is the number of tokens before the first token of q in block 'i'
# '(i * BM + BN - 1) // BN' is ceil(i * BM / BN)
for j in range((i * BM + BN - 1) // BN):
    # load k, v, and log_g for block 'j' from HBM, and calculate the indices of k and v
    real_bn = BN if (j + 1) * BN < s else s - j * BN
    indices_n      = j * BN + arange(0, BN)

    bk        = zeros(h, BN, d_k, dtype=bfloat16)
    bv        = zeros(h, BN, d_v, dtype=bfloat16)
    b_log_g_n = zeros(h, BN, d_v, dtype= float32)

    bk       [:,:real_bn].copy_(k    [:,j * BN: j * BN + real_bn])
    bv       [:,:real_bn].copy_(v    [:,j * BN: j * BN + real_bn])
    b_log_g_n[:,:real_bn].copy_(log_g[:,j * BN: j * BN + real_bn])

    # calculate the numerator of softmax(qk.T + mask) for block 'j'
    # only the tokens before the first token of q in block 'i' can be attended
    bqkT    = bmm(bq, bk.transpose(1, 2), float32) * sm_scale
    mask    = (indices_m[:,None] < s) & (indices_n[None,:] < i * BM)
    bqkT    = where(mask, bqkT, -inf)
    b_new_m = maximum(bm, amax(bqkT, dim=-1))
    scores  = exp(bqkT - b_new_m.view(h, BM, 1))

    # naively, this should be 'bv * exp(-b_log_g_n)'
    # however, this is numerically unstable, because the scale can be very large
    # instead, we use 'exp(b_log_g_m[:,:1] - b_log_g_n)'
    # because the index of b_log_g_m[:,0] is i * BM and 'indices_n[None,:] < i * BM',
    # all valid log_v_scale <= 0, there is no risk of dividing v by a near-zero value
    log_v_scale = minimum(b_log_g_m[:,:1] - b_log_g_n, tensor(0.))
    bv = (bv.to(float32) * exp(log_v_scale)).to(bfloat16)

    # update bm, bl, bo according to online softmax algorithm
    bo *= exp(bm - b_new_m).view(h, BM, 1)
    bl *= exp(bm - b_new_m)
    bo += bmm(scores.to(bv.dtype), bv, float32)
    bl += sum(scores, dim=-1)
    bm  = b_new_m

# naively, this should be 'bo * exp(b_log_g_m)'
# since the 'b_log_g_m[0]' is already used by all o vectors
# we use 'b_log_g_m - b_log_g_m[0]'
bo = bo * exp(b_log_g_m - b_log_g_m[:,:1])

bk = zeros(h, BM, d_k, dtype=bfloat16)
bv = zeros(h, BM, d_v, dtype=bfloat16)
bk[:,:real_bm].copy_(k[:,i * BM: i * BM + real_bm])
bv[:,:real_bm].copy_(v[:,i * BM: i * BM + real_bm])

bqkT    = bmm(bq, bk.transpose(1, 2), float32) * sm_scale
mask    = (indices_m[:,None] < s) & (indices_m[None,:] <= indices_m[:,None])
bqkT    = where(mask, bqkT, -inf)
b_new_m = maximum(bm, amax(bqkT, dim=-1))
scores  = exp(bqkT - b_new_m.view(h, BM, 1))

bo *= exp(bm - b_new_m).view(h, BM, 1)
bl *= exp(bm - b_new_m)
# because no intermediate log_g can be found, we resort to parallel form
# bo1 = s11 * (v1 / g1) * g1
# bo2 = s21 * (v1 / g1) * g2 + s22 * (v2 / g2) * g2
# bo3 = s31 * (v1 / g1) * g3 + s32 * (v2 / g2) * g3 + s33 * (v3 / g3) * g3
# ...
# thus, bo = s[:,1] * v1 * (g / g1) + s[:,2] * v2 * (g / g2) + ...
for j in range(BM):
    scale = exp(minimum(b_log_g_m - b_log_g_m[:,j:j+1], tensor(0.)))
    bo    += scores[:,:,j:j+1] * scale * bv[:,j:j+1].to(float32)
bl += sum(scores, dim=-1)
bm  = b_new_m

# normalized by the denominator of the softmax
bo = bo / bl.view(h, BM, 1)

# applying rms norm
b_rms = sqrt(mean(bo**2, dim=-1) + eps)
bo  = (bo * rmsnorm_weight.to(float32)) / b_rms.view(h, BM, 1)
```

```
        # cast to bfloat16 before writing back to HBM
        bo = bo.to(bfloat16)

        m  [:,i * BM: i * BM + real_bm].copy_(bm   [:,:real_bm])
        l  [:,i * BM: i * BM + real_bm].copy_(bl   [:,:real_bm])
        rms[:,i * BM: i * BM + real_bm].copy_(b_rms[:,:real_bm])
        o  [:,i * BM: i * BM + real_bm].copy_(bo   [:,:real_bm])

    return m, l, rms, o
```

Algorithm 4 presents one of the backward pass of Flash Fine-grained Attention. It calculates the backward pass of fused linear layer and RMSNorm and other statistics that is necessary for later usage.

---

**Algorithm 4** The Backward Pass of the Fused Linear Layer

---

```
def flash_fine_grained_attention_backward_linear(
    normed_o: Tensor,       #    [h, s, d_v], bfloat16
    d_final_o: Tensor,      #       [s, d_h], bfloat16
    linear_weight: Tensor,  # [d_h, h * d_v], bfloat16
    rmsnorm_weight: Tensor, #          [d_v], bfloat16
    rms: Tensor,            #          [h, s],  float32
):
    h, s, d_v = normed_o.shape
    rms_w = rmsnorm_weight.to(float32)

    do       = empty(h, s, d_v, dtype=bfloat16)
    d_weight = zeros(      d_v, dtype= float32)
    sum_sds  = empty(h, s,      dtype= float32)
    d_log_g  = empty(h, s, d_v, dtype= float32)

    # different heads are processes in parallel across different thread blocks
    for i_h in range(h):
        # backward of linear layer
        b_linear_weight = linear_weight[:,i_h * d_v: (i_h + 1) * d_v]
        bd_normed_o     = mm(d_final_o, b_linear_weight, float32)

        # backward of rmsnorm: input
        b_normed_o = normed_o[i_h].to(float32)
        rms_s      = rms[i_h].view(-1, 1)
        ydy        = b_normed_o * bd_normed_o
        c          = mean(ydy, dim=-1, keepdim=True)
        bdo        = (bd_normed_o * rms_w - (c * b_normed_o) / rms_w) / rms_s
        do[i_h]    = bdo.to(bfloat16)

        # backward of rmsnorm: weight
        bd_weight = sum(ydy / rms_w, dim=0)
        d_weight += bd_weight

        # backward of fine-grained attn:
        # row-wise sum of softmax(qk.T + mask) * d(softmax(qk.T + mask))
        # used later when calculating the gradient of q and k
        odo            = ydy - (c * b_normed_o**2) / (rms_w **2)
        b_sum_sds      = sum(odo, dim=-1)
        sum_sds[i_h] = b_sum_sds

        # backward of fine-grained attn: log_g through o
        bd_log_g      = odo
        d_log_g[i_h] = bd_log_g

    d_weight = d_weight.to(bfloat16)
    return do, d_weight, sum_sds, d_log_g
```

---

Algorithm 5 presents one of the backward pass of Flash Fine-grained Attention. It calculates the gradient with respect to $Q$.

---

**Algorithm 5** The Backward Pass of Flash Fine-grained Attention: $Q$

---

```
def flash_fine_grained_attention_backward_q(
    q: Tensor,       # [h, s, d_k], bfloat16
    k: Tensor,       # [h, s, d_k], bfloat16
```

```
    v: Tensor,         # [h, s, d_v], bfloat16
    log_g: Tensor,     # [h, s, d_v], bfloat16
    do: Tensor,        # [h, s, d_v], bfloat16
    sum_sds: Tensor,   # [h, s],      bfloat16
    m: Tensor,         # [h, s],      bfloat16
    l: Tensor,         # [h, s],      bfloat16
    sm_scale: float,
    BM: int = 64,
    BN: int = 64,
):
    # dimensions
    h, s, d_k = q.shape
    d_v = v.shape[-1]

    # final output
    dq = empty(h, s, d_k, dtype=bfloat16)

    # outer loop, executed in parallel across different thread blocks
    for i in range((s + BM - 1) // BM):
        real_bm = BM if (i + 1) * BM < s else s - i * BM
        indices_m        = i * BM + arange(0, BM)

        bdo       = zeros(h, BM, d_v, dtype=bfloat16)
        bq        = zeros(h, BM, d_k, dtype=bfloat16)
        b_log_g_m = zeros(h, BM, d_v, dtype= float32)
        b_sum_sds = zeros(h, BM,      dtype= float32)
        bm        = zeros(h, BM,      dtype= float32)
        bl        = zeros(h, BM,      dtype= float32)

        bdo       [:,:real_bm].copy_(do      [:,i * BM: i * BM + real_bm])
        bq        [:,:real_bm].copy_(q       [:,i * BM: i * BM + real_bm])
        b_log_g_m[:,:real_bm].copy_(log_g  [:,i * BM: i * BM + real_bm])
        b_sum_sds[:,:real_bm].copy_(sum_sds[:,i * BM: i * BM + real_bm])
        bm        [:,:real_bm].copy_(m       [:,i * BM: i * BM + real_bm])
        bl        [:,:real_bm].copy_(l       [:,i * BM: i * BM + real_bm])

        b_sum_sds = b_sum_sds.view(h, BM, 1)
        bm        = bm        .view(h, BM, 1)
        bl        = bl        .view(h, BM, 1)

        bdo_scaled = (bdo.to(float32) * exp(b_log_g_m - b_log_g_m[:,:1])).to(bfloat16)

        # accumulator of dq for block 'i'
        bdq = zeros([h, BM, d_k], dtype=float32)

        # non-diagonal blocks
        for j in range((i * BM + BN - 1) // BN):
            real_bn = BN if (j + 1) * BN < s else s - j * BN
            indices_n        = j * BN + arange(0, BN)

            bk        = zeros(h, BN, d_k, dtype=bfloat16)
            bv        = zeros(h, BN, d_v, dtype=bfloat16)
            b_log_g_n = zeros(h, BN, d_v, dtype= float32)

            bk        [:,:real_bn].copy_(k    [:,j * BN: j * BN + real_bn])
            bv        [:,:real_bn].copy_(v    [:,j * BN: j * BN + real_bn])
            b_log_g_n[:,:real_bn].copy_(log_g[:,j * BN: j * BN + real_bn])

            # recompute s = softmax(qk.T + mask)
            bqkT = bmm(bq, bk.transpose(1, 2), float32) * sm_scale
            mask = (indices_m[:,None] < s) & (indices_n[None,:] < i * BM)
            bqkT = where(mask, bqkT, -inf)
            bs   = exp(bqkT - bm) / bl

            # compute ds
            bv_scaled = (bv.to(float32) * exp(b_log_g_m[:,:1] - b_log_g_n)).to(bfloat16)
            bds       = bmm(bdo_scaled, bv_scaled.transpose(1, 2), float32)

            # compute dq
            bdqkT = (bs * (bds - b_sum_sds)) * sm_scale
            bdq += bmm(bdqkT.to(bfloat16), bk, float32)

        # diagonal block
        bk        = zeros(h, BM, d_k, dtype=bfloat16)
        bv        = zeros(h, BM, d_v, dtype=bfloat16)
        b_log_g_n = zeros(h, BM, d_v, dtype= float32)
```

```
bk        [:,:real_bm].copy_(k     [:,i * BM: i * BM + real_bm])
bv        [:,:real_bm].copy_(v     [:,i * BM: i * BM + real_bm])
b_log_g_n[:,:real_bm].copy_(log_g[:,i * BM: i * BM + real_bm])

bqkT = bmm(bq, bk.transpose(1, 2), float32) * sm_scale
mask = (indices_m[:,None] < s) & (indices_m[None,:] <= indices_m[:,None])
bqkT = where(mask, bqkT, -inf)
bs   = exp(bqkT - bm) / bl

# ds of the diagonal block can not be computed by matrix multiplication
bds = zeros(h, BM, BM, dtype=float32)
for j in range(BM):
    scale = exp(minimum(b_log_g_m - b_log_g_m[:,j:j+1], tensor(0.)))
    bds[:,:,j].copy_(sum(bdo.to(float32) * scale * bv[:,j:j+1].to(float32), dim=-1))

bdqkT = (bs * (bds - b_sum_sds)) * sm_scale
bdq  += bmm(bdqkT.to(bfloat16), bk, float32)

# write results back to HBM
bdq = bdq.to(bfloat16)
dq[:,i * BM: i * BM + real_bm].copy_(bdq[:,:real_bm])

return dq
```

Algorithm 6 presents one of the backward pass of Flash Fine-grained Attention. It calculates the gradient with respect to $K, V$, and $\log \Gamma$.

---

**Algorithm 6** The Backward Pass of Flash Fine-grained Attention: $K, V$, and $\log \Gamma$

```
def flash_fine_grained_attention_backward_kv(
    q: Tensor,        # [h, s, d_k], bfloat16
    k: Tensor,        # [h, s, d_k], bfloat16
    v: Tensor,        # [h, s, d_v], bfloat16
    log_g: Tensor,    # [h, s, d_v], bfloat16
    do: Tensor,       # [h, s, d_v], bfloat16
    sum_sds: Tensor,  # [h, s],      bfloat16
    m: Tensor,        # [h, s],      bfloat16
    l: Tensor,        # [h, s],      bfloat16
    sm_scale: float,
    BM: int = 64,
    BN: int = 64,
):
    # dimensions
    h, s, d_k = q.shape
    d_v = v.shape[-1]

    # final outputs
    dk      = empty(h, s, d_k, dtype=bfloat16)
    dv      = empty(h, s, d_v, dtype=bfloat16)
    d_log_g = empty(h, s, d_v, dtype= float32)

    # outer loop, executed in parallel across different thread blocks
    for j in range((s + BN - 1) // BN):
        real_bn = BN if (j + 1) * BN < s else s - j * BN
        indices_n          = j * BN + arange(0, BN)

        bk        = zeros(h, BN, d_k, dtype=bfloat16)
        bv        = zeros(h, BN, d_v, dtype=bfloat16)
        b_log_g_n = zeros(h, BN, d_v, dtype= float32)

        bk        [:,:real_bn].copy_(k     [:,j * BN: j * BN + real_bn])
        bv        [:,:real_bn].copy_(v     [:,j * BN: j * BN + real_bn])
        b_log_g_n[:,:real_bn].copy_(log_g[:,j * BN: j * BN + real_bn])

        bv_scaled = (bv.to(float32) * exp(b_log_g_n[:,-1:] - b_log_g_n)).to(bfloat16)

        # accumulator of dk and dv for block 'j'
        bdk = zeros([h, BN, d_k], dtype=float32)
        bdv = zeros([h, BN, d_v], dtype=float32)

        # non-diagonal blocks
        # '(j + 1) * BN' is the index of the first token after block 'j'
```

```
# thus, '((j + 1) * BN) // BM' is the first non-diagonal block of q and o
begin = ((j + 1) * BN) // BM
end = (s + BM - 1) // BM
for i in range(min(begin, end), end):
    real_bm = BM if (i + 1) * BM < s else s - i * BM
    indices_m       = i * BM + arange(0, BM)

    bdo       = zeros(h, BM, d_v, dtype=bfloat16)
    bq        = zeros(h, BM, d_k, dtype=bfloat16)
    b_log_g_m = zeros(h, BM, d_v, dtype= float32)
    b_sum_sds = zeros(h, BM,      dtype= float32)
    bm        = zeros(h, BM,      dtype= float32)
    bl        =  ones(h, BM,      dtype= float32)

    bdo       [:,:real_bm].copy_(do    [:,i * BM: i * BM + real_bm])
    bq        [:,:real_bm].copy_(q     [:,i * BM: i * BM + real_bm])
    b_log_g_m[:,:real_bm].copy_(log_g  [:,i * BM: i * BM + real_bm])
    b_sum_sds[:,:real_bm].copy_(sum_sds[:,i * BM: i * BM + real_bm])
    bm        [:,:real_bm].copy_(m     [:,i * BM: i * BM + real_bm])
    bl        [:,:real_bm].copy_(l     [:,i * BM: i * BM + real_bm])

    b_sum_sds = b_sum_sds.view(h, BM, 1)
    bm        = bm        .view(h, BM, 1)
    bl        = bl        .view(h, BM, 1)

    # recompute s = softmax(qk.T + mask)
    bqkT = bmm(bq, bk.transpose(1, 2), float32) * sm_scale
    mask = ((indices_m < s) & (indices_m >= (j + 1) * BN))[:,None]
    bqkT = where(mask, bqkT, -inf)
    bs   = exp(bqkT - bm) / bl

    # compute ds and dv
    bdo_scaled = (bdo.to(float32) * exp(b_log_g_m - b_log_g_n[:,-1:])).to(bfloat16)
    bdv       += bmm(bs.to(bfloat16).transpose(1, 2), bdo_scaled)
    bds        = bmm(bdo_scaled, bv_scaled.transpose(1, 2), float32)

    # compute dk
    bdqkT = (bs * (bds - b_sum_sds)) * sm_scale
    bdk  += bmm(bdqkT.to(bfloat16).transpose(1, 2), bq, float32)

bdv *= exp(b_log_g_n[:,-1:] - b_log_g_n)

# diagonal block
bdo       = zeros(h, BN, d_v, dtype=bfloat16)
bq        = zeros(h, BN, d_k, dtype=bfloat16)
b_log_g_m = zeros(h, BN, d_v, dtype= float32)
b_sum_sds = zeros(h, BN,      dtype= float32)
bm        = zeros(h, BN,      dtype= float32)
bl        =  ones(h, BN,      dtype= float32)

bdo       [:,:real_bn].copy_(do    [:,j * BN: j * BN + real_bn])
bq        [:,:real_bn].copy_(q     [:,j * BN: j * BN + real_bn])
b_log_g_m[:,:real_bn].copy_(log_g  [:,j * BN: j * BN + real_bn])
b_sum_sds[:,:real_bn].copy_(sum_sds[:,j * BN: j * BN + real_bn])
bm        [:,:real_bn].copy_(m     [:,j * BN: j * BN + real_bn])
bl        [:,:real_bn].copy_(l     [:,j * BN: j * BN + real_bn])

b_sum_sds = b_sum_sds.view(h, BN, 1)
bm        = bm        .view(h, BN, 1)
bl        = bl        .view(h, BN, 1)

bqkT = bmm(bq, bk.transpose(1, 2), float32) * sm_scale
mask = (indices_n[:,None] < s) & (indices_n[None,:] <= indices_n[:,None])
bqkT = where(mask, bqkT, -inf)
bs   = exp(bqkT - bm) / bl

# ds and dv of the diagonal block can not be computed by matrix multiplication
bds = zeros(h, BN,  BN, dtype=float32)
for i in range(BN):
    scale = exp(minimum(b_log_g_n[:,i:i+1] - b_log_g_n, tensor(0.)))
    bdoi  = bdo[:,i:i+1].to(float32)
    bdv  += bs[:,i:i+1].transpose(1, 2) * scale * bdoi
    bds[:,i].copy_(sum(bdoi * scale * bv.to(float32), dim=-1))

bdqkT = (bs * (bds - b_sum_sds)) * sm_scale
bdk  += bmm(bdqkT.to(bfloat16).transpose(1, 2), bq, float32)
```

```
    # calculate d_log_g through v
    bd_log_g = -bv.to(float32) * bdv

    # write results back to HBM
    bdk = bdk.to(bfloat16)
    bdv = bdv.to(bfloat16)

    dk      [:,j * BN: j * BN + real_bn].copy_(bdk      [:,:real_bn])
    dv      [:,j * BN: j * BN + real_bn].copy_(bdv      [:,:real_bn])
    d_log_g[:,j * BN: j * BN + real_bn].copy_(bd_log_g[:,:real_bn])

return dk, dv, d_log_g
```

Algorithm 7 shows how to combine the previous kernels together to form the Flash Fine-grained Attention.

---

**Algorithm 7** Flash Fine-grained Attention

---

```
class FusedFineGrainedAttentionNormLinear(Function):
    @staticmethod
    def forward(
        ctx: FunctionCtx,
        q: Tensor,                #     [h, s, d_k], bfloat16
        k: Tensor,                #     [h, s, d_k], bfloat16
        v: Tensor,                #     [h, s, d_v], bfloat16
        log_g: Tensor,            #     [h, s, d_v],  float32
        rmsnorm_weight: Tensor,   #          [d_v], bfloat16
        linear_weight: Tensor,    # [d_h, h * d_v], bfloat16
        sm_scale: float,
        eps: float,
    ):
        m, l, rms, normed_o = flash_fine_grained_attention_forward(
            q, k, v, log_g, rmsnorm_weight, sm_scale, eps)
        normed_o_input = rearrange(normed_o, 'h s d -> s (h d)')
        final_o = linear(normed_o_input, linear_weight)
        ctx.save_for_backward(
            q, k, v, log_g, rmsnorm_weight, linear_weight, m, l, rms, normed_o)
        ctx.sm_scale = sm_scale
        return final_o

    @staticmethod
    def backward(
        ctx: FunctionCtx,
        d_final_o: Tensor, # [s, d_h], bfloat16
    ):
        q, k, v, log_g, rms_w, linear_w, m, l, rms, normed_o = ctx.saved_tensors
        sm_scale = ctx.sm_scale
        normed_o_input = rearrange(normed_o, 'h s d -> s (h d)')
        d_linear_w = mm(d_final_o.T, normed_o_input, float32)
        do, d_rms_w, sum_sds, d_log_g_o = flash_fine_grained_attention_backward_linear(
                normed_o, d_final_o, linear_w, rms_w, rms)
        dq = flash_fine_grained_attention_backward_q(
            q, k, v, log_g, do, sum_sds, m, l, sm_scale)
        dk, dv, d_log_g_v = flash_fine_grained_attention_backward_kv(
            q, k, v, log_g, do, sum_sds, m, l, sm_scale)
        d_log_g = d_log_g_o + d_log_g_v
        return dq, dk, dv, d_log_g, d_rms_w, d_linear_w, None, None
```

---

