# OpenReview forum: "FiX: Introducing Fine-grained Forget Gate into Softmax Attention"
_ICML.cc/2026/Conference — ICML 2026 regular_

### Official Review · Reviewer_ymKA · 2026-02-24

**Soundness:** 3
**Presentation:** 3
**Significance:** 3
**Originality:** 3
**Overall Recommendation:** 5
**Confidence:** 3

**Summary:**

The paper presents a practical recipe (FiX) to realize fine-grained, per-channel forgetting inside softmax attention by exploiting RMSNorm’s normalizing effect and implementing VO-decay with many engineering safeguards. The conceptual link to FoX is neat and the infra work is thorough. Experiments on medium-scale LMs and long-context tasks show certain gains versus baselines.

**Compliance With Llm Reviewing Policy:**

Affirmed.

**Final Justification:**

The paper proposes FiX, which realizes fine-grained, per-channel forgetting inside softmax attention by leveraging RMSNorm’s normalizing effect and implementing VO-decay with engineering safeguards. The strengths of this work include a clear motivation, low latency overhead, and thorough engineering and implementation efforts. The main weakness is that the performance improvement is relatively modest. In the rebuttal, the authors note that they are currently exploring a Hybrid FiX + FoX approach, which can achieve further gains—under gated attention, the improvement of Hybrid FiX + FoX over FoX is roughly comparable to FoX over RoPE (an increase of ~8e-3 in training loss). Overall, I find this work interesting and could be promising, and I have therefore increased my rating from 4 (Weak Accept) to 5 (Accept).

**Key Questions For Authors:**

1. I find it confusing for the claim in Eq.(12) that RMSNorm($o_i$) = RMSNorm($o_i'$) even if $\epsilon=0$. Please provide a more detailed explanations. Additionally, the authors claim that "the attention output is always RMS-normalized before subsequent operations", yet in the common pre-layer RMSNorm-style LM, the output is first added with the residual, and then normalized before FFN. Is it contradictory to the authors' claim?
2. Why is the RMSNorm in Eq.(14)  applied to $x_i W_f^{↓}$ rather than to the whole expression? Can the authors provide some explanations or ablations towards this formulation?
3. Gated-attention variants are much easier to implement, stable, and do not require fused kernels or paged VF cache. Can the authors provide a direct comparison between FiX and gated-attention baselines (including gated + simple long-context techniques)?
4. Can the authors report measured inference metrics such as peak GPU memory and decoding latency to highlight the effecacy of the proposed infra work?

**Limitations:**

1. The method requires many low-level engineering and the observed gains relative to FoX seem modest, which raises concerns about whether the engineering investment pays off broadly.
2. The paper claims FiX helps long-context extrapolation; however, Section 5.3 does not provide a tight algorithmic explanation of why FiX should consistently help length extrapolation beyond empirical curves. A stronger theoretical or diagnostic argument would be valuable.
3. Many essential algorithmic and experimental details such as the explicit form of Eq.(14) gating parameterization and the exact model sizes and total training token budgets  are found only in the appendix. It would be better if the authors can move these details to the main text so readers can judge and understand better.

**Strengths And Weaknesses:**

Strength:
1. The motivation is clear. The paper makes a compelling case to try to realize FoX-style fine-grained forgetting inside softmax attention based on the observation of certain equivalance of QK-FoX and VO-FoX under RMSNorm.
2. The engineering and implementation work is thorough. The authors pay substantial attention to numerical stability and efficiency and propose low-rank parametrization, Mamba-style parameterization for the gate, FP32 fused RMSNorm, rescale / blocking to keep GEMM efficiency, and a paged VF cache to limit inference memory. These are nontrivial and necessary for a practical system.

Weakness:
1. The method requires many low-level optimizations. This raises questions about how easily FiX can be adopted in common open-source stacks without custom kernels.
2. The absolute improvements reported appear relatively small compared to FoX: e.g., in Fig.2 the FiX / FiX-RoPE training loss improvements over FoX-RoPE after 48B tokens are on the order of a few 1e-3 (≈3e-3 / 7e-3), and on other metrics the gains are often modest.

---

> ### Author Rebuttal · Authors · 2026-03-30
>
> We thank the reviewer for the thorough review. All additional results are in https://anonymous.4open.science/api/repo/icml_rebuttal-6DE2/file/4.pdf.
>
> **1. Engineering complexity (W1 & L1).**
>
> Our engineering complexity is comparable to standard FlashAttention, which already implements the four operators (1 fwd and 3 bwd) that we provide in the appendix. FiX only makes targeted modifications to these operators. We note that linear attention variants such as Gated DeltaNet[1] and Kimi Delta Attention[2] have similar or greater engineering complexity, yet have been successfully validated at scale in Qwen 3.5[3] and Kimi Linear[2], respectively.
>
> **2. Clarification of Eq. 12 and relations with PreNorm.**
>
> From Eq. 9 and 10, $\boldsymbol{o}_i$ and $\boldsymbol{o}'_i$ share the same numerator $\boldsymbol{x}$ but differ by a positive scalar denominator, so $\boldsymbol{o}_i = \alpha\boldsymbol{x}$, $\boldsymbol{o}'_i = \beta\boldsymbol{x}$ with $\alpha,\beta>0$. Since RMSNorm is scale-invariant when $\epsilon=0$:
>
> $$\text{RMSNorm}(c\boldsymbol{x})_k = \frac{c x_k w_k}{c\sqrt{\frac{1}{n}\sum_j x_j^2}} = \frac{x_k w_k}{\sqrt{\frac{1}{n}\sum_j x_j^2}}$$
>
> the positive scalar cancels, giving $\text{RMSNorm}(\boldsymbol{o}_i) = \text{RMSNorm}(\boldsymbol{o}'_i)$.
>
> The RMSNorm we refer to is *inside* $\text{Attn}(\cdot)$, not the pre/post-layer norm. Concretely, using PreNorm: $\boldsymbol{h}_{l+1} = \boldsymbol{h}_l + \text{Attn}(\text{RMSNorm}(\boldsymbol{h}_l))$, where the input $\boldsymbol{x} = \text{RMSNorm}(\boldsymbol{h}_l)$ applies the pre-layer norm and $\text{Attn}$ internally computes $\text{Attn}(\boldsymbol{x}) = \boldsymbol{W}_o \text{RMSNorm}(\boldsymbol{O})$ as its output. This internal RMSNorm on the attention output is standard in architectures like FoX (see their Table 3 showing its benefit), Mamba2[4], and Qwen3.5[3] and is independent of whether PreNorm or PostNorm is used at the layer level.
>
> **3. Why RMSNorm on the gate's low-rank latent (Eq. 14)?**
>
> This follows standard practice for low-rank projections. For example, DeepSeek's MLA[5] applies RMSNorm on the low-rank latent of compressed queries before the up-projection. The RMSNorm stabilizes the intermediate representation and prevents magnitude drift through the bottleneck. This is not specific to our method.
>
> **4. Comparison with gated attention baselines.**
>
> FiX is orthogonal to gated attention and can be used together. We conducted additional experiments comparing FoX-RoPE+Gated vs. FiX-RoPE+Gated at the 760M scale. FiX-RoPE+Gated achieves lower training loss than FoX-RoPE+Gated, with an improvement consistent with the FiX-RoPE vs FoX-RoPE gap reported in the paper (see Figure 1 in the link). This confirms that FiX provides complementary gains on top of gated attention.
>
> **5. Peak GPU memory and decoding latency.**
>
> See Table 1 in the link. FiX's inference overhead is minimal: comparable latency and decode rate to RoPE, with only 4% more peak memory.
>
> **6. Gains relative to FoX and engineering investment. (also W2)**
>
> The relative improvement of FoX to RoPE $L_\text{RoPE}-L_\text{FoX}$ is around $8 \times 10^{-3}$. Meanwhile, the relative improvement of FiX compared to FoX $L_\text{FoX}-L_\text{FiX}$ is around $5 \times 10^{-3}$, which is comparable to FoX's gain over RoPE. *On top of this, RoPE improves FiX and FoX equally.* Moreover, we conducted preliminary experiments on a **hybrid FiX+FoX** architecture, which achieves further training loss improvements (see Figure 2 in the link). This suggests FiX opens a complementary dimension for enhancing softmax attention, justifying the engineering investment.
>
> **7. Why FiX helps length extrapolation.**
>
> In each attention block, the computation path is $v$ -> attn\_output -> residual. Different channels of $v$ exhibit different drift rates along the sequence dimension. FoX applies uniform scalar decay across all channels, while FiX applies per-channel decay, allowing each dimension to be controlled independently. This reduces the overall standard deviation drift in the residual stream across sequence positions. Our analysis (see Figure 3 in the link) confirms that FiX has the smallest feature stddev drift compared to FoX and RoPE, following the analytical framework of [6] which uses stddev drift to explain length generalization in recurrent models.
>
> **8. Key details in appendix.**
>
> We appreciate this suggestion. In a future version, we will move the requested components into the main text to improve readability.
>
> [1] Gated Delta Networks: Improving Mamba2 with Delta Rule, ICLR 2025
>
> [2] Kimi Linear: An Expressive, Efficient Attention Architecture, arxiv
>
> [3] https://qwen.ai/blog?id=qwen3.5
>
> [4] Transformers are SSMs: Generalized Models and Efficient Algorithms Through Structured State Space Duality, ICML 2024
>
> [5] DeepSeek-V2: A Strong, Economical, and Efficient Mixture-of-Experts Language Model, arxiv
>
> [6] Understanding and Improving Length Generalization in Recurrent Models, ICML 2025

---

> > ### Author Rebuttal · Reviewer_ymKA · 2026-04-01
> >
> > Thanks for the detailed and thorough response. I appreciate the clarifications on the derivations (especially Eq. 12) and the additional experiments. Overall, the rebuttal addresses my concerns well and improves the clarity and completeness of the work.
> >
> > While the absolute improvement of FiX still seems relatively modest, I think that this direction is interesting and the method is reasonably well-motivated. As a suggestion, it would be more solid to further explore hybrid designs such as FiX+FoX in more depth, as the preliminary results you provided indicate that this direction could lead to additional gains.
> >
> > Based on the rebuttal, I decide to raise my score.

---

> > > ### Author Response · Authors · 2026-04-02
> > >
> > > We sincerely thank the reviewer for the thoughtful re-evaluation and for raising the score.
> > >
> > > We are glad that our clarifications have adequately addressed the reviewer's concerns.
> > > We also appreciate the reviewer's recognition for the potential of our architecture.
> > >
> > > Thank you again for the constructive engagement throughout the review process, which has meaningfully improved the paper.

---

### Official Review · Reviewer_CDWZ · 2026-03-11

**Soundness:** 3
**Presentation:** 2
**Significance:** 3
**Originality:** 3
**Overall Recommendation:** 4
**Confidence:** 4

**Summary:**

FIX propose novel method to introduce gating mechanism in the softmax attention.

**Compliance With Llm Reviewing Policy:**

Affirmed.

**Final Justification:**

I acknowledge all authors' rebuttal, and I have a better understanding than before. I want to finalize score.

**Key Questions For Authors:**

- Can you insert intuitive concept figure in the main body?

**Limitations:**

- Limited comparison with baselines
  - I think interpolating RoPE methods should be in the comparison. Why the gating mechanims is required, if existing method is enouhgly good?
- Limited analysis in the efficiency. No latency reports, no nsight analysis results.

**Strengths And Weaknesses:**

FIX provide indepth analysis of the proposed gating algorithm, and emphirical result on a realistic benchmark set and model scale.

---

> ### Author Rebuttal · Authors · 2026-03-30
>
> We thank the reviewer for the positive assessment and constructive suggestions. We address each point below.
>
> **1. Intuitive concept figure.**
>
> We have prepared a concept figure along with detailed description illustrating FiX's architecture and how fine-grained forget gate is applied on the VO side. Please see the [concept figure](https://anonymous.4open.science/api/repo/icml_rebuttal-6DE2/file/concept.pdf).
>
> **2. Comparison with RoPE interpolation methods (e.g., YaRN).**
>
> RoPE interpolation methods such as YaRN[1] are designed to extend context length by **fine-tuning** a pretrained model with modified positional encodings. In contrast, our length extrapolation experiments (Section 5.3) evaluate whether models can ***natively*** generalize to longer contexts without any fine-tuning or modification. These are fundamentally different evaluation settings: YaRN requires additional fine-tuning to adapt the model, while we test the inherent length generalization capability of each architecture. Therefore, a direct comparison between FiX and YaRN would not be meaningful in our experimental setup. Our additional experiment shows that directly applying YaRN to RoPE models without fine-tuning will hurt the performance inside the training context length (see [figure](https://anonymous.4open.science/api/repo/icml_rebuttal-6DE2/file/yarn.pdf)). In addition, we note that FiX is fully compatible with RoPE (as shown by FiX-RoPE in our experiments), and similar interpolation methods could also be applied on top.
>
> Moreover, the ability to generalize beyond training context length is only an extra bonus of FiX. The main aim of FiX is still to outperform our baselines inside the training context length on training loss and downstream tasks.
>
> **3. Efficiency analysis: latency, memory, and throughput.**
>
> See [inference metrics](https://anonymous.4open.science/api/repo/icml_rebuttal-6DE2/file/inference.pdf). FiX achieves inference latency and decode throughput comparable to RoPE, with only 4% more peak memory. This demonstrates that the engineering optimizations (Flash Fine-grained Attention, paged VF cache) effectively amortize the overhead of fine-grained forget gate.
>
> [1] YaRN: Efficient Context Window Extension of Large Language Models, ICLR 2024

---

> > ### Author Rebuttal · Reviewer_CDWZ · 2026-04-05
> >
> > Authors resolved my concerns. I will keep my scores.

---

> > > ### Author Response · Authors · 2026-04-07
> > >
> > > We sincerely thank the reviewer for the positive assessment and for confirming that our responses have resolved the concerns.
> > >
> > > We are grateful for the reviewer's constructive suggestions, especially the concept figure and the YaRN comparison, which helped us improve the clarity and completeness of our paper.
> > >
> > > Thank you again for the valuable feedback that has improved the presentation of our paper.

---

### Official Review · Reviewer_GJsV · 2026-03-12

**Soundness:** 3
**Presentation:** 3
**Significance:** 3
**Originality:** 3
**Overall Recommendation:** 4
**Confidence:** 3

**Summary:**

This paper proposes a new architecture called FiX, aiming to successfully incorporate the fine-grained (element-wise) forgetting gate into the causal softmax attention mechanism that is widely used in modern large language models. The author decouples the forgetting mechanism and transforms it into applying cumulative element-wise multiplications directly on the V and O vectors. And in the engineering aspect, Flash Fine-grained Attention was designed and pageVF Cache was proposed to balance the pressure on memory and bandwidth. The model based on the proposed technical framework was verified in practice and outperformed the comparison methods with better performance.

**Compliance With Llm Reviewing Policy:**

Affirmed.

**Final Justification:**

The author addressed most of my concerns. I still maintain the original positive score.

**Key Questions For Authors:**

1. Force the ε value of RMSNorm to be set to the minimum value of 1e-30. Is this merely a way to cover up a mathematical flaw? Moreover, if future architectures no longer use RMSNorm, will the entire method then fail?
2. Why does the first layer rely on the embedding table instead of a linear projection, merely to prevent gradient explosion? Wouldn't this inconsistent design hinder the model from handling continuous non-textual data such as images or audio?
3. Even though the Paged VF cache saves memory space by storing the product of floating-point numbers, does this result in a slower memory bandwidth during the decoding process? Moreover, why didn't the authors provide performance test results that demonstrate the actual token generation rate?

**Limitations:**

The author may need to discuss the specificities of the current approach in large language models.

**Strengths And Weaknesses:**

Strength
Soundness:
This manuscript demonstrates that under the RMSNorm, the Q-K and V-O gating methods are mathematically equivalent. It also addresses the numerical underflow issue that arises during training by using a fused 32-bit floating-point operator and scaling techniques.
Presentation:
This manuscript clearly explains the transition process from linear attention to the proposed method, and includes detailed pseudo-code for forward propagation and backward propagation to facilitate reproduction.
Significance:
This fine-grained forgetting gate technology enhances the long-context modeling capability, and the proposed page-based virtual cache mechanism can be easily integrated into modern reasoning engines like vLLM.
Originality:
Applying position encoding and gate encoding directly to the V-O vector, rather than in the usual Q-K space, is an innovative approach.
Weakness:
Soundness:
Setting the ε value of RMSNorm to an extremely small 1e-30 may lead to issues regarding the stability of training in different hardware environments or with mixed precision settings.
Significance:
These experiments only tested models with no more than 7.6M parameters. Therefore, it is not clear whether this method can be applied to modern models with over 7B parameters.

---

> ### Author Rebuttal · Authors · 2026-03-30
>
> We thank the reviewer for the detailed and constructive feedback, and for recognizing FiX's soundness, clarity, and originality. We address each concern below.
>
> **1.  The setting of $\epsilon$ in RMSNorm and what if future architectures no longer use RMSNorm.**
>
> *Setting $\epsilon=10^{-30}$: is this merely covering up a mathematical flaw?*
>
> No, quite the opposite. We use $\epsilon=10^{-30}$ to *highlight* our theoretical contribution. Eq. 12 states that $\text{RMSNorm}(\boldsymbol{o}') = \text{RMSNorm}(\boldsymbol{o})$ when $\epsilon=0$. We deliberately use an extreme value ($10^{-30}$) compared with the conventional value ($10^{-6}$, see Figure 5 in the paper) to demonstrate, both theoretically and empirically, that our contribution is not merely proposing FiX's mathematical form, but rigorously analyzing *why* it works under RMSNorm.
>
> Practically, the setting of $\epsilon$ is not strictly constrained to 1e-30. Our new experiments show that $\epsilon=10^{-20}$ or $10^{-15}$ yields **identical** training loss to $\epsilon=10^{-30}$ (see [ablation of eps](https://anonymous.4open.science/api/repo/icml_rebuttal-6DE2/file/eps.pdf)). This is consistent with DeepSeek mHC[1], which validates that $\epsilon$ values such as $10^{-20}$ work stably even at 27B scale.
>
> *If future architectures no longer use RMSNorm, will FiX fail?*
>
> There are three cases to consider:
>
> (a) If RMSNorm is replaced by another normalization (e.g., LayerNorm), the equivalence between QK-FoX and VO-FoX can be similarly proved under these norms, since the key property is scale invariance. See response 2 to reviewer 4.
>
> (b) If the discussion is about pre-norm vs. post-norm placement (e.g., replacing pre-layer RMSNorm with other schemes), this is orthogonal to FiX. See response 2 to reviewer 4.
>
> (c) As evidenced by many works such as Mamba2[2], FoX, Qwen3.5[3], using RMSNorm on attention output is empirically beneficial. As such, one can always apply an RMSNorm after the attention output.
>
> Thus, this is not a limitation to FiX.
>
> **2. First-layer forget gate embedding: why not a linear projection? Does it hinder multi-modal use?**
>
> The embedding design addresses a fundamental difficulty: learning a reasonable projection layer for discrete embedding inputs. In practice, the L2 norm of the forget gate projection's gradient in the first layer is unbounded, causing training instability. The embedding avoids this by directly learning per-token forget gates.
>
> Regarding multimodal applicability: the use of embedding design will not hinder multi-modal capability, and many practices have been demonstrated to successfully adapt embedding table to multi-modal inputs, such as Google's Gemma 3n[4] and VQ-VAE[5]. Specifically, Google's Gemma 3n integrates Per-Layer Embedding into a multimodal architecture (text + image + audio) by appending the same embedding for all non-text inputs. And VQ-VAE applies VQ to discretize the contiguous vector inputs.  Meanwhile, as a fallback, one can replace the first layer with FoX. [Our exploration](https://anonymous.4open.science/api/repo/icml_rebuttal-6DE2/file/first_fox.pdf) shows this achieves comparable or even slightly better performance than full FiX.
>
> **3. Does the paged VF cache cause slower decoding?**
>
> No, our proposed Paged VF cache instead accelerates decoding compared with naive VF cache precisely because it saves memory space and decoding is memory bound. We discussed 3 kinds of strategies in our paper (Section 4.3) in detail. In short, naive VF cache needs 2 times memory access compared to standard KV cache because forget gates are stored in fp32 and no F cache requires additional V cache write back. While paged VF cache only needs an extra of 1/page_size memory access compared to standard KV cache. This means the token generation rate should be similar to RoPE and FoX, which is also justified by our new experiments (see our [inference metrics](https://anonymous.4open.science/api/repo/icml_rebuttal-6DE2/file/inference.pdf)). FiX achieves decode throughput comparable to RoPE (16.3 vs 16.1 tok/s) with only 4% more peak memory. The paged VF cache effectively amortizes the cost of maintaining the forget gate cache.
>
> **4. 760M model scale and specificities of FiX in large language models.**
>
> The 760M scale is the standard evaluation setting adopted by closely related works including FoX and Titans[6]. We acknowledge larger-scale experiments would be valuable but currently lack the computational resources. Notably, FiX's infrastructure (fused kernels, paged VF cache, vLLM compatibility) was specifically designed with scalability in mind.
>
> [1] mHC: Manifold-Constrained Hyper-Connections, arxiv
>
> [2] Transformers are SSMs: Generalized Models and Efficient Algorithms Through Structured State Space Duality, ICML 2024
>
> [3] https://qwen.ai/blog?id=qwen3.5
>
> [4] https://ai.google.dev/gemma/docs/gemma-3n
>
> [5] Neural Discrete Representation Learning, NeurIPS 2017
>
> [6] Titans: Learning to Memorize at Test Time, NeurIPS 2025

---

> > ### Author Rebuttal · Reviewer_GJsV · 2026-04-01
> >
> > The authors have successfully addressed most of the issues I raised.

---

> > > ### Author Response · Authors · 2026-04-02
> > >
> > > We sincerely thank the reviewer for confirming that our responses have adequately addressed the concerns.
> > >
> > > We are grateful for the reviewer's recognition of FiX's soundness, clarity, and originality. In particular, the reviewer's question inspired us to explore the hybrid FiX+FoX design, which shows promising potential to further improve performance. We plan to investigate this direction more thoroughly in future work.
> > >
> > > Thank you again for the constructive feedback, which has meaningfully improved the paper.

---

### Official Review · Reviewer_egou · 2026-03-13

**Soundness:** 3
**Presentation:** 2
**Significance:** 1
**Originality:** 2
**Overall Recommendation:** 3
**Confidence:** 3

**Summary:**

This paper look at how to add a fine-grained forget gate to standard softmax attention. To do this, they shift the forget mechanism from the scalar attention-logit  to the value/output side, allowing forgetting to happen at the hidden channel level rather than token level. Based on this, this paper proposes FiX and evaluates it against FoX-style/RoPE baselines on language modeling, long-context extrapolation and commonsense reasoning.

**Compliance With Llm Reviewing Policy:**

Affirmed.

**Final Justification:**

The scaling argument is indirect and requires empirical validation at larger scales, and FiX introduces non-trivial overhead in training time, memory, and complexity that is not sufficiently justified by the current gains. Therefore, I will maintain my current score.

**Key Questions For Authors:**

see weakness.

**Limitations:**

see weakness.

**Strengths And Weaknesses:**

Strengths.

- The fused attention–RMSNorm–linear kernel (preserving float32 where needed) and the paged VF cache are systems-level contributions.

- The motivation is sound.

- Experiments included training loss, LM perplexity, general downstream tasks, and long-context extrapolation. Key design choices are all ablated.


Weaknesses.

- The method relies on an extremely small RMSNorm epsilon (1e-30), which may severely influence training stability, mixed-precision robustness, limited it's practicality. A formulation that does not rely on such an extreme setting would be much more convincing.

- Experiments are based on a small moderate model size (~760M) with one training setup. It is unclear whether the method works equally well on larger models, other architectures, or more modern LLM recipes.

- While the engineering complexity is high, the actual gains is small. For instance, the average gain over fox is limited, FiX-RoPE is even worse than FoX-RoPE in many tasks (table 1). The complexity-benefit tradeoff is not very compelling.


- A detailed analysis of parameter overhead, wall-clock time, and memory usage compare to baselines are needed.

---

> ### Author Rebuttal · Authors · 2026-03-30
>
> We thank the reviewer for recognizing the systems-level contributions and thorough ablation study of FiX. We address each concern below.
>
> **1. RMSNorm epsilon (1e-30) and training stability.**
>
> The choice of $\epsilon=10^{-30}$ is intentionally conservative to emphasize the theoretical equivalence (Eq. 12). In practice, much larger values work equally well. We conducted additional experiments comparing $\epsilon=10^{-20}$ or $10^{-15}$ vs. $10^{-30}$ and found the training loss curves are **identical** (see [ablation of eps](https://anonymous.4open.science/api/repo/icml_rebuttal-6DE2/file/eps.pdf)). Note that this setting is also consistent with industrial practices, such as the DeepSeek mHC paper [1], which demonstrates that $\epsilon$ values much smaller than the conventional $10^{-5}$ or $10^{-6}$ (e.g., $10^{-20}$) are viable even at 27B scale with stable training. Therefore, the $\epsilon$ requirement does not impose practical stability risk.
>
> Furthermore, for mixed-precision scenarios, we designed a Fused-RMSNorm operator (Section 4.2 in the paper) to calculate the denominator of RMSNorm in fp32 format to improve the training stability. This operator can be used normally in mixed-precision training. In fact, all the experiments in the paper are conducted under a bf16-fp32 mixed-precision setting, where bf16 is used for activations and fp32 for accumulations, aligned with the common practice in SOTA open source LLMs[2, 3]. This also demonstrates the effectiveness of our fused kernel even with $\epsilon=10^{-30}$.
>
> **2. Model scale limited to ~760M.**
>
> We follow the standard evaluation setting (760M) adopted by closely related works including FoX and Titans [4]. We acknowledge that larger-scale validation would strengthen the paper, but we currently lack the computational resources for such experiments. We note that FiX's design — fused kernels, paged VF cache, compatibility with vLLM — was engineered with scalability in mind, and we plan to validate at larger scales as resources become available.
>
> **3. Gains over FoX are small; FiX-RoPE is worse than FoX-RoPE on some tasks.**
>
> The relative improvement of FoX to RoPE $L_\text{RoPE}-L_\text{FoX}$ is around $8 \times 10^{-3}$. Meanwhile, the relative improvement of FiX compared to FoX $L_\text{FoX}-L_\text{FiX}$ is around $5 \times 10^{-3}$, which is comparable to FoX's gain over RoPE. *On top of this, RoPE improves FiX and FoX equally.* For downstream tasks, individual task variance is expected. FoX-RoPE also does not outperform RoPE on every single benchmark. The important metric is overall performance: FiX achieves the best average (55.24) and geometric mean (53.79) across all 10 metrics in Table 1, outperforming all baselines.
>
> Additionally, we attempted preliminary exploration on a **hybrid FiX+FoX architecture** that uses FiX in some layers and FoX in others. The results show further improved training loss over standalone FiX (see [use fox instead of forget embeddings](https://anonymous.4open.science/api/repo/icml_rebuttal-6DE2/file/first_fox.pdf)), suggesting that FiX opens a complementary axis of improvement for softmax attention.
>
> **4. Parameter overhead, wall-clock time, and memory analysis.**
>
> Detailed analysis of overhead is provided below:
>
> **Parameter overhead:** The 767M backbone is shared across all models. FiX uses a similar structure to RoPE and FoX, and only introduces \~11.5M additional parameters for low-rank projection of fine-grained forget gates. Specifically, the FiX in our main experiments  has 36 layers. In the first layer, FiX introduces a forget gate embedding of $32000\times1280 = 41\text{M}$ parameters, where $32000$ is the vocabulary size and $1280$ is the dimension of fine-grained forget gates of all heads. Critically, this embedding is only needed during training and can be offloaded to CPU memory during inference, adding zero GPU overhead. All other 35 layers use low-rank projection consisting of a down and up projection between hidden states and forget gates. The dimensions of the hidden states, low-rank latent, and the forget gates are 1280, 128, 1280 respectively. So the total added parameters are $35 \times (1280\times128 + 128\times1280) = 11.5M$.
>
> **Efficiency and Memory usage:** We conduct additional experiments to investigate runtime cost of FiX (see [inference metrics](https://anonymous.4open.science/api/repo/icml_rebuttal-6DE2/file/inference.pdf) for results). Specifically, FiX achieves inference latency and decode throughput comparable to RoPE, with only 4% more peak memory, demonstrating that the engineering optimizations (Flash Fine-grained Attention, paged VF cache) effectively amortize the overhead of fine-grained forget gate.
>
> [1] mHC: Manifold-Constrained Hyper-Connections, arxiv
>
> [2] Kimi K2: Open Agentic Intelligence, arxiv
>
> [3] https://qwen.ai/blog?id=qwen3.5
>
> [4] Titans: Learning to Memorize at Test Time, NeurIPS 2025

---

> > ### Author Rebuttal · Reviewer_egou · 2026-04-03
> >
> > Thank you for the rebuttal. However, my main concerns remain unresolved. It is still unclear whether the method generalizes beyond the single moderate-scale (~760M) setting to larger models, other architectures, or more modern LLM training recipes. In addition, the rebuttal does not directly address training-time or training-memory overhead. Finally, I am not convinced that the gains justify the additional complexity. Therefore, I will keep my original score.

---

> > > ### Author Response · Authors · 2026-04-07
> > >
> > > We thank the reviewer for the continued engagement. We respectfully address the remaining concerns below.
> > >
> > > **1. Model scale.**
> > >
> > > FiX's architectural modification over FoX is analogous to how Kimi Delta Attention (KDA)[1] modifies Gated DeltaNet (GDN)[2]. Both FiX and KDA introduce fine-grained, per-channel gating on top of a scalar-gated baseline. KDA has been successfully scaled to 27B parameters in Kimi Linear[1], demonstrating that this type of architectural modification scales well in practice.
> > >
> > > **2. Other architectures.**
> > >
> > > The architecture used in our experiments is a modern Transformer that incorporates SwiGLU, short convolution, QK-Norm, and O-Norm. These techniques are used because they are verified to improve Transformer performance and many of which have been successfully scaled to very large scale (e.g., Qwen3.5[3], Gemma4[4]). Furthermore, we have verified FiX's improvement both with and without gated attention (see [results with gated attention](https://anonymous.4open.science/api/repo/icml_reply-6EA8/file/gate.pdf) for the results with gated attention), demonstrating FiX's adaptability across different architectural configurations.
> > >
> > > **3. Training recipes.**
> > >
> > > Our training recipe in the paper directly follows the setting of FoX (hyperparameters) and Gated DeltaNet[2] (dataset). We have additionally tested FiX on the high-quality ClimbMix[5] dataset published by Nvidia (see [ClimbMix results](https://anonymous.4open.science/api/repo/icml_reply-6EA8/file/climbmix.pdf)). Notably, **the improvement of FiX over FoX (${\sim}1.4 \times 10^{-2}$) is larger than the improvement of FoX over RoPE (${\sim}0.8 \times 10^{-2}$) on ClimbMix dataset, while the improvement of FoX over RoPE is basically the same between ClimbMix and FineWeb-Edu (both $\sim 0.8 \times 10^{-2}$)**. Given that the dataset is the most important part of a training recipe, our results demonstrate that FiX's gains generalize across recipes, and that FiX tends to yield larger improvements on higher-quality datasets.
> > >
> > > **4. Training time and memory overhead.**
> > >
> > > We provide the training overhead comparison below:
> > >
> > > |  | Training Time (h) | Memory Usage (GB) |
> > > |---|---|---|
> > > | RoPE | 46.0 | 62.6 |
> > > | FoX | 56.4 | 62.5 |
> > > | FiX | 65.5 | 77.0 |
> > >
> > > This is the training overhead on 760M-parameter transformers trained on 48B tokens using 8x RTX Pro 6000 GPUs. FoX adds 23% wall-clock time over RoPE. Notably, FiX only adds 16% extra training time over FoX, while potentially achieving a comparable or larger improvement in training loss. The memory overhead is primarily due to forget gate activations retained for backpropagation. It can be reduced via gradient checkpointing or recomputation. We highlight two additional points:
> > >
> > > (a) With the rise of agentic applications, inference efficiency becomes increasingly more important than training cost. It is worthwhile to invest more training time to improve model quality at the same inference efficiency. As shown in our [inference metrics](https://anonymous.4open.science/api/repo/icml_rebuttal-6DE2/file/inference.pdf), FiX achieves inference latency and decode throughput comparable to RoPE with only 4% more peak memory.
> > >
> > > (b) The training overhead stems from FiX's element-wise and matrix multiplication operations both competing for register resources on our RTX Pro 6000 GPUs. These GPUs lack the next-generation matmul instructions available on Hopper (wgmma) and Blackwell (tcgen05) architectures, which significantly reduce register pressure. On these newer architectures, FiX has substantial potential for improved training efficiency.
> > >
> > > [1] Kimi Linear: An Expressive, Efficient Attention Architecture, arxiv
> > >
> > > [2] Gated Delta Networks: Improving Mamba2 with Delta Rule, ICLR 2025
> > >
> > > [3] https://qwen.ai/blog?id=qwen3.5
> > >
> > > [4] https://deepmind.google/models/gemma/gemma-4
> > >
> > > [5] Nemotron-CLIMB: CLustering-based Iterative Data Mixture Bootstrapping for Language Model Pre-training, NeurIPS 2025

---

### Decision · Program_Chairs · 2026-04-30

**Decision:**

Accept (regular)

**Comment:**

The paper proposes FiX, a nontrivial generalization of FoX that enables vector-valued forgetting in softmax attention. It observes that a value-output-side reformulation of FoX is equivalent to standard FoX after immediate RMSNorm while making elementwise gating feasible. The implementation involves substantial systems work, including custom fused kernels and cache design. The gains are modest but generally consistent. Reviewers appreciated the technical motivation and engineering depth, with remaining concerns on whether the added complexity and overhead are sufficiently justified without larger-scale validation.